# Local angiogenic interplay of Vegfc/d and Vegfa controls brain region-specific emergence of fenestrated capillaries

**Sweta Parab**[1,2], **Olivia A Card**[1,2], **Qiyu Chen**[3], **Michelle America**[4], **Luke D Buck**[1,2], **Rachael E Quick**[1,2], **William F Horrigan**[1,2], **Gil Levkowitz**[3], **Benoit Vanhollebeke**[4], **Ryota L Matsuoka**[1,2]*

[1]Department of Neurosciences, Lerner Research Institute, Cleveland Clinic, Cleveland, United States; [2]Department of Molecular Medicine, Cleveland Clinic Lerner College of Medicine, Case Western Reserve University, Cleveland, United States; [3]Departments of Molecular Cell Biology and Molecular Neuroscience, The Weizmann Institute of Science, Rehovot, Israel; [4]Laboratory of Neurovascular Signaling, Department of Molecular Biology, ULB Neuroscience Institute, Université libre de Bruxelles, Gosselies, Belgium

**\*For correspondence:** matsuor@ccf.org

**Abstract** Fenestrated and blood-brain barrier (BBB)-forming endothelial cells constitute major brain capillaries, and this vascular heterogeneity is crucial for region-specific neural function and brain homeostasis. How these capillary types emerge in a brain region-specific manner and subsequently establish intra-brain vascular heterogeneity remains unclear. Here, we performed a comparative analysis of vascularization across the zebrafish choroid plexuses (CPs), circumventricular organs (CVOs), and retinal choroid, and show common angiogenic mechanisms critical for fenestrated brain capillary formation. We found that zebrafish deficient for Gpr124, Reck, or Wnt7aa exhibit severely impaired BBB angiogenesis without any apparent defect in fenestrated capillary formation in the CPs, CVOs, and retinal choroid. Conversely, genetic loss of various Vegf combinations caused significant disruptions in Wnt7/Gpr124/Reck signaling-independent vascularization of these organs. The phenotypic variation and specificity revealed heterogeneous endothelial requirements for Vegfs-dependent angiogenesis during CP and CVO vascularization, identifying unexpected interplay of Vegfc/d and Vegfa in this process. Mechanistically, expression analysis and paracrine activity-deficient *vegfc* mutant characterization suggest that endothelial cells and non-neuronal specialized cell types present in the CPs and CVOs are major sources of Vegfs responsible for regionally restricted angiogenic interplay. Thus, brain region-specific presentations and interplay of Vegfc/d and Vegfa control emergence of fenestrated capillaries, providing insight into the mechanisms driving intra-brain vascular heterogeneity and fenestrated vessel formation in other organs.

## Editor's evaluation

This study presents a fundamental approach to understanding genetic redundancy of Vegf family, Wnt7, Gpr124 and Reck in mediating zebrafish blood brain barrier angiogenesis and formation of region-specific fenestrated capillaries. The work comprehensively examines interplay of Vegfc/d and Vegfa and their sources for regionally-restricted angiogenesis. The study provides detail on the region-specific expression of Vegf ligands, and assesses the permeability of the fenstrated capillaries using dye injections into the vasculature.

## Introduction

Throughout the body, the vascular system delivers oxygen, nutrients, and hormones while removing metabolic wastes. In addition to this general function, each organ acquires vascular properties unique to its function during development, leading to the generation of vascular phenotypic heterogeneity in the organism (*Aird, 2007*). This inter-organ vascular heterogeneity is an evolutionarily conserved feature of the endothelium across diverse vertebrate species (*Yano et al., 2007*) and is important to maintain homeostasis in the organism. Recent bulk and single-cell RNA sequencing (scRNA-seq) in humans and mice has further validated significant transcriptional heterogeneity of endothelial cells across various organs (*Sabbagh et al., 2018*; *Jambusaria et al., 2020*; *Nolan et al., 2013*; *Kalucka et al., 2020*) and even within the same tissue (*Kalucka et al., 2020*; *Matsuoka et al., 2022*; *Vanlandewijck et al., 2018*). How this inter- and intra-organ endothelial heterogeneity is established and maintained remains an outstanding question in vascular biology and organ physiology.

The brain exhibits striking intra-organ endothelial phenotypic heterogeneity (*Kalucka et al., 2020*; *Vanlandewijck et al., 2018*; *Winkler et al., 2022*; *Garcia et al., 2022*). In most regions of the brain, endothelial cells form the semi-permeable blood-brain barrier (BBB) (*Langen et al., 2019*; *Daneman and Prat, 2015*), while those in the restricted regions of the brain develop highly permeable pores, or fenestrae (*Matsuoka et al., 2022*; *Miyata, 2015*). The BBB is an integral part of the neurovascular unit and forms a brain-vascular interface vital for maintaining the optimal chemical and cellular milieu of the brain (*Langen et al., 2019*; *Daneman and Prat, 2015*). In contrast, the choroid plexuses (CPs) and circumventricular organs (CVOs) are midline brain ventricular structures vascularized with fenestrated capillaries that lack the BBB (*Matsuoka et al., 2022*; *Miyata, 2015*). Fenestrated vasculature in these organs allows for high vascular permeability, serving to maintain fluid balance and mediate neuroendocrine activities (*Matsuoka et al., 2022*; *Miyata, 2015*). Thus, intra-brain vascular heterogeneity is fundamental to brain homeostasis and brain region-specific neural function. Additionally, in steady states, the BBB limits immune cell infiltration from the circulatory system into the brain parenchyma (*Daneman and Prat, 2015*; *Daneman et al., 2010*), while fenestrated capillaries are implicated to allow peripheral immune cells to enter the CPs for immune surveillance (*Shipley et al., 2020*; *Reboldi et al., 2009*; *Ghersi-Egea et al., 2018*; *Llovera et al., 2017*; *Szmydynger-Chodobska et al., 2009*). However, in neuroinflammatory conditions where BBB breakdown occurs, immune cells can penetrate the brain parenchyma through vasculature that loses tight barrier properties, exacerbating neuroinflammatory responses. Thus, heterogeneous vascular barrier integrity is an important regulator of neural immune responses associated with age, injury, and/or disease. Despite the physiological and clinical importance, the mechanisms underlying intra-brain vascular heterogeneity establishment remain unclear. Moreover, the cellular and molecular basis of fenestrated brain capillary development remains poorly understood.

We recently reported that a unique set of angiogenic cues (Vegfab, Vegfc, and Vegfd) is required for fenestrated vessel formation in the zebrafish myelencephalic CP, an equivalent to the fourth ventricular CP in mammals (*Parab et al., 2021*). Intriguingly, the combined loss of these Vegfs has little impact on the formation of neighboring brain vasculature which displays BBB molecular signatures (*Parab et al., 2021*), indicating that endothelial cell-type-specific angiogenesis is a key mechanism driving local vascular heterogeneity within the brain. To further explore this possibility, we expanded our analysis to different CPs and CVOs in the present study to address the following questions. First, we asked whether Wnt/β-catenin signaling, a critical signaling pathway for BBB vascular development (*Langen et al., 2019*; *Daneman and Prat, 2015*), plays a role in fenestrated brain vessel formation. Second, we investigated whether the combination of the Vegf ligands identified in our previous study is a universal angiogenic inducer of fenestrated vasculature across the central nervous system (CNS). Our combined analysis revealed fenestrated capillary type-selective angiogenic mechanisms, their local and inter-tissue heterogeneity, and a previously unappreciated function of Vegfc/d in this process.

## Results

### Genetic loss of Wnt/β-catenin signaling leads to severely impaired angiogenesis in the brain parenchyma without any apparent defect in CPs vasculature

Wnt/β-catenin signaling is well established as a central regulator of brain angiogenesis and BBB formation/maintenance (*Langen et al., 2019*; *Daneman and Prat, 2015*). We first asked whether this signaling pathway is required for fenestrated vascular development across the CNS. We employed the zebrafish model due to its ex utero animal development, rapid formation of brain vascular heterogeneity, and its facile visualization in whole-brain tissues of live animals. For example, 3D confocal imaging of the endothelial cell-specific transgenic (Tg) zebrafish reporter *Tg(kdrl:ras-mCherry)* carrying the *Et(cp:EGFP)* enhancer trap line allowed us to simultaneously visualize brain vasculature and two anatomically separate CPs, the diencephalic and myelencephalic CP (dCP and mCP), which are equivalent to the third and fourth ventricular CP in mammals, respectively (*Figure 1A and A'*; *Bill and Korzh, 2014*; *García-Lecea et al., 2008*; *Henson et al., 2014*).

Zebrafish carrying double transgenic *Tg(plvap:EGFP);Tg(glut1b:mCherry)* reporters (*Umans et al., 2017*; *Fetsko et al., 2023*) enabled us to map transcriptionally heterogeneous networks of brain and meningeal vasculature at 120 hours post fertilization (hpf) (*Figure 1B and B'*). Specifically, three major vascular phenotypes were observed at this stage of development and onward: (1) blood vessels exhibiting high levels of *Tg(plvap:*EGFP), but low levels of *Tg(glut1b:*mCherry), expression; (2) those exhibiting low levels of *Tg(plvap:*EGFP), but high levels of *Tg(glut1b:*mCherry), expression; and (3) those with low/medium levels of both *Tg(plvap:*EGFP) and *Tg(glut1b:*mCherry) expression. Previous studies showed that the *Tg(plvap:EGFP)* reporter labels immature brain endothelial cells during embryogenesis (*Umans et al., 2017*; *Fetsko et al., 2023*). However, this *Tg(plvap:*EGFP) expression is diminished between 3 and 5 days post fertilization (dpf) (*Umans et al., 2017*; *Fetsko et al., 2023*) in the brain regions where they differentiate into the BBB phenotype characterized by strong expression of established BBB markers, including Glut1 (*Umans et al., 2017*; *Fetsko et al., 2023*) and the multidrug resistance efflux pump P-glycoprotein (Pgp) (*Umans and Taylor, 2012*; *Schumacher and Mollgård, 1997*). Indeed, we observed at 10 dpf that strong *Tg(glut1b:*mCherry)[+] brain and meningeal vasculature exhibited high levels of Pgp and Glut1 immunoreactivity and faint *Tg(plvap:*EGFP) expression (*Figure 1C–C''', E and G*; *Figure 1—figure supplement 1A–D*). In contrast, blood vessels formed in tissues around the CPs retained strong *Tg(plvap:*EGFP) expression and displayed undetectable levels of Glut1 and Pgp immunoreactivity even at 10 dpf (*Figure 1C–C''', D and F*; *Figure 1—figure supplement 1A–C*). Combined with the absence of the BBB tight junction marker Claudin-5 expression in these strong *Tg(plvap:*EGFP)[+] vessels at both the transcriptional (*van Leeuwen et al., 2018*) and protein (*Parab et al., 2021*) levels, this molecular marker analysis suggests that zebrafish CP vasculature exhibits fenestrated endothelial molecular signatures characterized by both strong Plvap and low levels of BBB marker expression reported in the adult mouse CP (*Wang et al., 2019*).

To investigate the role of Wnt/β-catenin signaling in fenestrated brain vessel formation, we chose to analyze zebrafish that harbor mutations in *gpr124*, *reck*, or *wnt7aa*. Current evidence indicates that Gpr124 and Reck act as receptor co-factors of Wnt7a/b-specific canonical β-catenin pathway (*Cho et al., 2017*; *Vanhollebeke et al., 2015*). Both *gpr124[s984]* and *wnt7aa[ulb2]* zebrafish mutants carry out-of-frame mutations that induce a premature stop codon leading to truncated proteins that lack functional domains (*Vanhollebeke et al., 2015*; *Martin et al., 2022*). The newly generated *reck[ulb3]* zebrafish mutants harbor a 12 bp in-frame deletion in the third cysteine knot motif (*Figure 1—figure supplement 2*), close to the binding motifs of Reck to Gpr124 and Wnt7a/b that are critical for brain angiogenesis activity. We observed that *gpr124[s984]*, *reck[ulb3]*, and *wnt7aa[ulb2]* mutants all displayed severe angiogenesis defects in the brain parenchyma (*Figure 1I–L'*). These phenotypes are consistent with previous reports in zebrafish carrying mutations in each of these genes (*Umans et al., 2017*; *Fetsko et al., 2023*; *Vanhollebeke et al., 2015*; *Martin et al., 2022*; *Ulrich et al., 2016*; *America et al., 2022*) and also similar to those documented in mouse knockouts (*Daneman et al., 2009*; *Kuhnert et al., 2010*; *Stenman et al., 2008*; *Cho et al., 2017*), showing the high conservation of this signaling pathway in brain angiogenesis between zebrafish and mammals. However, despite these severe defects in brain parenchymal angiogenesis, we found that vascularization of both the dCP and mCP were not compromised in any of these mutants (*Figure 1H and M–S*).

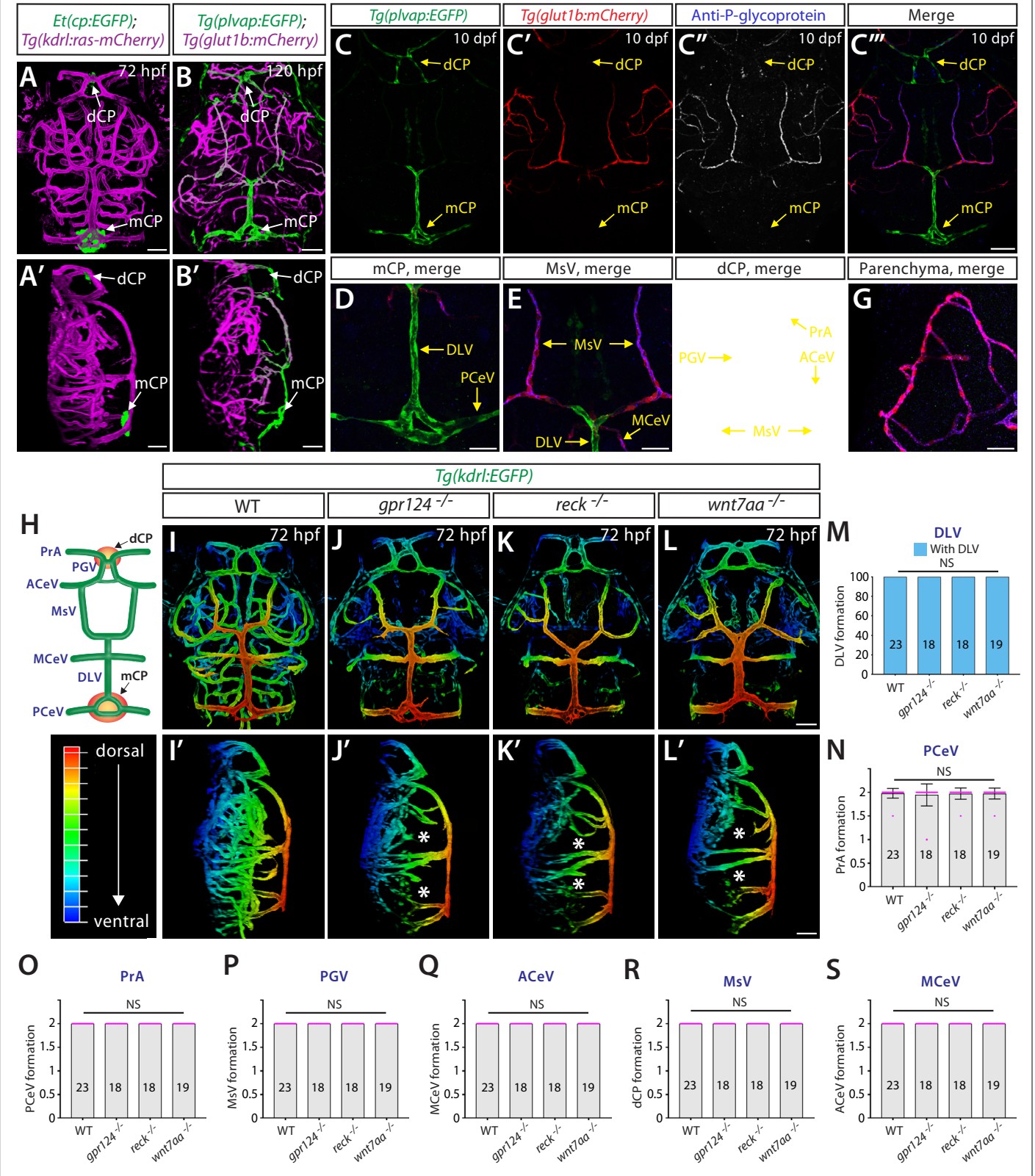

**Figure 1.** Genetic loss of Wnt/β-catenin signaling leads to severely impaired angiogenesis in the brain parenchyma without any apparent defect in fenestrated choroid plexus (CP) vasculature. (**A, A'**) Dorsal (**A**) and lateral (**A'**) views of a 72 hours post fertilization (hpf) *Et(cp:EGFP);Tg(kdrl:ras-mCherry)* larval head point to the locations of the diencephalic and myelencephalic CP (dCP and mCP, respectively). (**B, B'**) Dorsal (**B**) and lateral (**B'**) views of a 72 hpf *Tg(plvap:EGFP);Tg(glut1b:mCherry)* larval head indicate fenestrated and blood-brain barrier (BBB) states of meningeal and brain

*Figure 1 continued on next page*

*Figure 1 continued*

vasculature. Blood vessels formed in the dCP and mCP display strong *Tg(plvap:*EGFP) expression. (**C–C″′**) Dorsal views of the 10 days post fertilization (dpf) *Tg(plvap:EGFP);Tg(glut1b:mCherry)* larval head immunostained for P-glycoprotein (Pgp), an endothelial marker for the BBB state. Vasculature which forms in the dCP and mCP shows strong *Tg(plvap:*EGFP) expression and absent expression of both *Tg(glut1b:*mCherry) transgene and Pgp immunoreactivity. (**D–G**) Magnified, merged images of the immunostained larva shown in (**C–C″′**). In support of *Tg(plvap:*EGFP) and *Tg(glut1b:*mCherry) transgene expression, Pgp immunoreactivity was detected in *Tg(glut1b:*mCherry)+ blood vessels, including the MsV, MCeV, and blood vessels in the midbrain parenchyma. *Tg(plvap:*EGFP)+ blood vessels were devoid of Pgp immunoreactivity. (**H**) Schematic diagram of the dorsal view of cranial vasculature at around 72–10 dpf, illustrating the locations of the two CPs and distinct cranial blood vessels used for quantifications. **PrA**: prosencephalic artery, **PGV**: pineal gland vessel, **ACeV**: anterior cerebral vein, **MsV**: mesencephalic vein, **MCeV**: middle cerebral vein, **DLV**: dorsal longitudinal vein, **PCeV**: posterior cerebral vein. (**I–L′**) Dorsal (**I–L**) and lateral (**I′–L′**) views of 72 hpf wild-type (WT) (**I, I′**), *gpr124*-/- (**J, J′**), *reck*-/- (**K, K′**), and *wnt7aa*-/- (**L, L′**) larval head vasculature visualized by *Tg(kdrl:*EGFP) expression. Color-coded maximum projection images indicate the most dorsal vessels in red and ventral ones in blue with a gradual color shift from dorsal to ventral (the color codes are shown in a panel left to **I′**). Asterisks indicate severely impaired angiogenesis in the brain parenchyma of these mutants compared to WT (**I′–L′**). (**M–S**) Quantification of vessel formation in the dorsal meningeal and brain compartments at 72 hpf (the number of animals examined per genotype is listed in the panel). No significant difference was detected in *gpr124*-/-, *reck*-/-, or *wnt7aa*-/- larvae compared to WT. Data are means ± SD. NS: not significant. For panels **N–S**, each data point shown in magenta represents individual animal's vessel formation score. Scale bars: 50 μm in (**A–B′**), in (**C″′**) for (**C–C″′**), in (**D–G**), in (**L**) for (**I–L**), and in (**L′**) for (**I′–L′**).

The online version of this article includes the following figure supplement(s) for figure 1:

**Figure supplement 1.** Characterization of endothelial marker and β-catenin reporter expression in zebrafish larval brain vasculature.

**Figure supplement 2.** Characterization of the *reck^ulb3* allele.

## Wnt/β-catenin signaling deficiency does not cause any apparent vessel formation defect in the neurohypophysis and organum vasculosum of the lamina terminalis

Next, we examined vascular formation in the neurohypophysis (NH, posterior pituitary) and the organum vasculosum of the lamina terminalis (OVLT) (*Figure 2A*), a secretory and sensory CVO, respectively (*Miyata, 2015*). Fenestrated capillaries were either previously shown (*Gordon et al., 2019*; *Anbalagan et al., 2018*) or anticipated to form (*García-Lecea et al., 2017*) in these brain regions. In support of these previous reports, we noted strong *Tg(plvap:*EGFP) expression and low levels of Glut1 and Pgp immunoreactivity in vasculature formed around the NH and OVLT at 120 hpf and onward (*Figure 2B–D″*; *Figure 1—figure supplement 1E–F″*). We observed that none of the *gpr124*, *reck*, or *wnt7aa* mutant larvae displayed an apparent vessel defect in these brain regions (*Figure 2E–K*), indicating that fenestrated vessel formation does not rely on Wnt/β-catenin signaling across the brain.

To examine if blood vessels in these CVOs and CPs indeed exhibit higher vascular permeability than those forming the BBB, we injected fluorescent dextran dyes of different molecular weights (3 kDa and 10 kDa) into the bloodstream at 6 dpf (*Figure 2L*). At this developmental stage, functional BBB is already established in the midbrain and hindbrain (*O'Brown et al., 2019*). We detected limited dye accumulation in the midbrain parenchyma following both the 3 kDa and 10 kDa tracer injections (*Figure 2M and Q*), as previously reported (*O'Brown et al., 2019*). In contrast, substantially increased accumulation of both tracers was detected in brain regions around the dCP, NH, and OVLT, with 3 kDa dextran displaying a slightly higher degree of accumulation (*Figure 2N–Q*). These observations are consistent with the previous tracer permeability studies conducted in adult and larval zebrafish (*Henson et al., 2014*; *Gordon et al., 2019*; *Anbalagan et al., 2018*; *Jeong et al., 2008*). Furthermore, we detected low levels of β-catenin activity in the vascular endothelium around the CPs, NH, and OVLT at 6 dpf, as compared to more prominent β-catenin activity in BBB-forming central arteries of the hindbrain (*Figure 1—figure supplement 1G–K*). Together, these combined molecular and functional characterizations suggest that strong *Tg(plvap:*EGFP)+ CP and CVO vasculature comprises permeable capillaries with fenestrated endothelial molecular characteristics (*Wang et al., 2019*) at around 5 dpf and later in development.

## Developmental angiogenesis at the interface of the dCP and pineal gland

We next sought to explore angiogenic cues responsible for vascularization of the fenestrated brain vascular beds that is independent of Wnt/β-catenin signaling. Previous studies, including our recent work, characterized the vascularization processes of the mCP (*Parab et al., 2021*; *Bill and Korzh,*

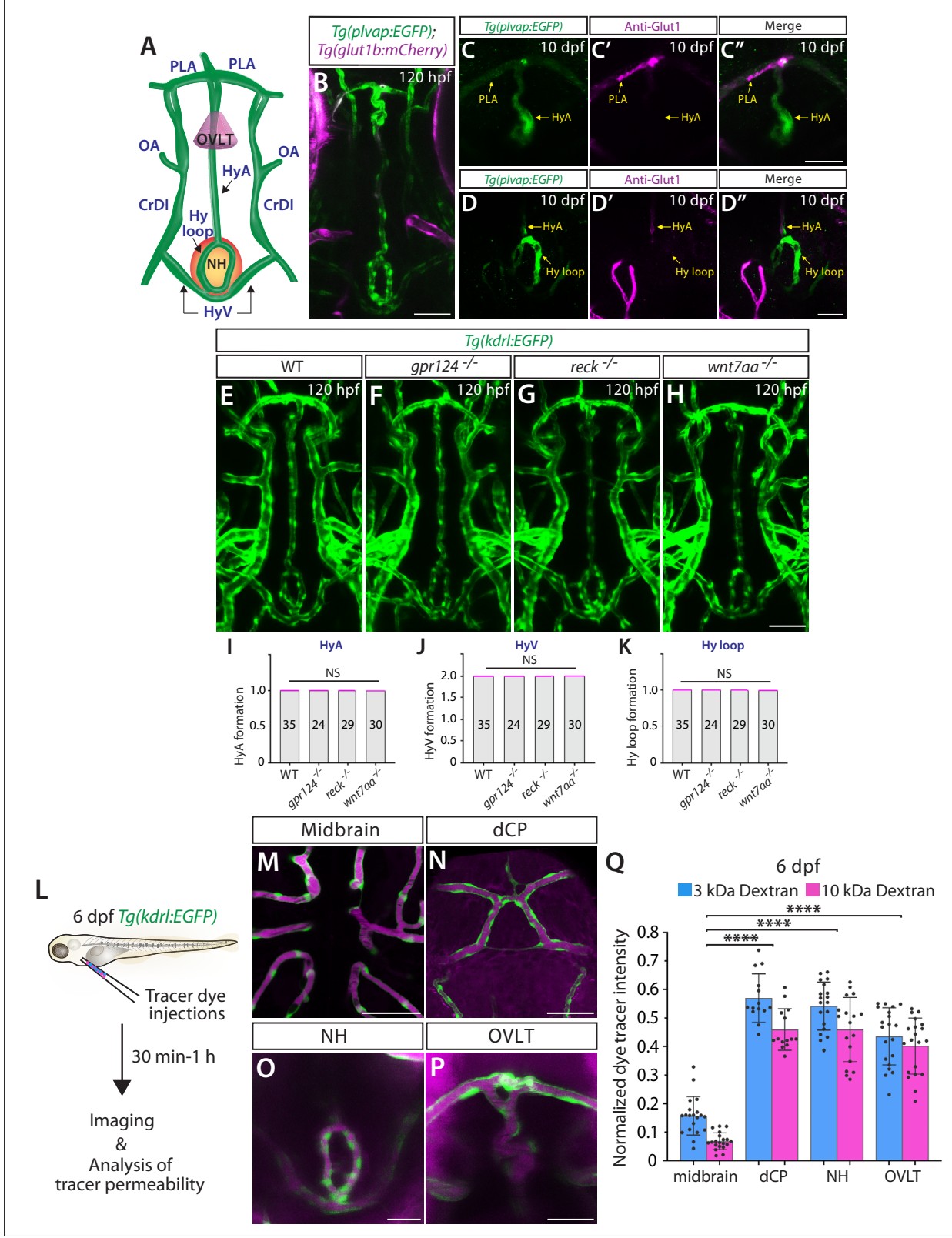

**Figure 2.** Wnt/β-catenin signaling deficiency does not cause any apparent defect in fenestrated capillary formation in the neurohypophysis (NH) and organum vasculosum of the lamina terminalis (OVLT). (**A**) Schematic diagram of vasculature in the ventral brain at around 5–10 days post fertilization (dpf), illustrating the locations of the OVLT, the NH, and distinct blood vessels used for quantifications. **HyA**: hypophyseal artery, **HyV**: hypophyseal veins, **Hy loop**: hypophyseal loop, **PLA**: palatocerebral arteries, **OA**: optic artery, **CrDI**: cranial division of the internal carotid artery. (**B**) Dorsal view of

*Figure 2 continued on next page*

Figure 2 continued

120 hours post fertilization (hpf) Tg(plvap:EGFP);Tg(glut1b:mCherry) ventral brain shows strong Tg(plvap:EGFP) expression in the Hy loop, HyA, and PLA compared to its fainter signals in the HyV. (C–D") Dorsal views of the 10 dpf Tg(plvap:EGFP) head immunostained for Glut1. Glut1 immunoreactivity was undetectable in a rostral portion of the Tg(plvap:EGFP)⁺ HyA that lies in proximity to the OVLT (C–C") and in the Tg(plvap:EGFP)⁺ Hy loop (D–D"). Faint signals were detected in a caudal portion of the HyA that resides close to the Hy loop (D–D"). (E–H) Dorsal views of 120 hpf wild-type (WT) (E), gpr124⁻/⁻ (F), reck⁻/⁻ (G), and wnt7aa⁻/⁻ (H) ventral brain vasculature visualized by Tg(kdrl:EGFP) expression. gpr124⁻/⁻, reck⁻/⁻, or wnt7aa⁻/⁻ larvae formed vasculature in the NH/OVLT regions similar to WT. (I–K) Quantification of ventral brain vessel formation at 120 hpf (the number of animals examined per genotype is listed in the panel). No significant difference was detected in gpr124⁻/⁻, reck⁻/⁻, or wnt7aa⁻/⁻ larvae compared to WT. Each data point shown in magenta represents individual animal's vessel formation score. (L) Experimental workflow of tracer dye injections and subsequent imaging and tracer permeability analysis. Tg(kdrl:EGFP) larvae at 6 dpf were co-injected with 3 kDa and 10 kDa dextran dyes conjugated with different fluorophores. (M–P) Merged images of 3 kDa tetramethylrhodamine-conjugated dextran dye (magenta) and Tg(kdrl:EGFP)⁺ vasculature at 6 dpf. Unlike the midbrain parenchyma (M) where a functional blood-brain barrier (BBB) is established, a higher amount of tracer accumulation was detected in tissues around the diencephalic choroid plexus (dCP), NH, and OVLT brain regions (N–P). (Q) Quantification of normalized tracer intensity across the different brain regions at 6 dpf reveals a significant increase in tracer accumulation around the dCP, NH, and OVLT brain regions compared to the midbrain parenchyma. **** indicates p<0.0001 by one-way analysis of variance (ANOVA) followed by Tukey's HSD test. Statistical significance was calculated for each dye tracer across different brain regions and represents differences in the graph. Scale bars: 50 µm in (B), in (H) for (E–H), in (M–N); 25 µm in (C") for (C–C"), in (D") for (D–D"), in (O–P).

2014; Henson et al., 2014), however, no angiogenic cues have been reported for dCP vascularization. Intriguingly, through our literature reviews and immunostaining of Tg(kdrl:ras-mCherry);Et(cp:EGFP) larvae with an antibody for rhodopsin, a marker for pineal photoreceptor cells (Laurà et al., 2012), we found that the dCP and pineal gland (PG) lie adjacent to strong Tg(plvap:EGFP)⁺ vasculature (Figure 3A–D; Video 1). The PG is the major neuroendocrine organ that secretes melatonin, which controls circadian rhythms and sleep-wake cycles (Turek and Gillette, 2004). Identification of this unique dCP/PG vascular interface motivated us to determine its angiogenic mechanisms. The Et(cp:EGFP)⁺ cells in the dCP were outlined by Claudin-5 tight junction protein expression, a marker for CP epithelial cells (Henson et al., 2014; van Leeuwen et al., 2018), allowing us to visualize dCP epithelial cells by immunostaining for Claudin-5 without the Et(cp:EGFP) reporter (Figure 3E–E").

Angiogenic steps leading to vascularization of the dCP/PG interface have not been well characterized. We noted that this vascularization process initiates at very early embryonic stages. By 23 hpf, the anterior cerebral vein (ACeV) sprouts bilaterally from the primordial hindbrain channel and extends dorsally to start forming the bilateral prosencephalic artery (PrA) (Figure 3F). The PrA then extends anteroventrally to connect with the cranial division of the internal carotid artery around 26 hpf (Figure 3G). Anastomosis of this vascular plexus occurs by 32 hpf (Figure 3H) after the systemic circulation of blood begins at approximately 24–26 hpf (Isogai et al., 2001). Through 72 hpf, the vascular structure undergoes remodeling and maturation to establish a functional circuit (Figure 3I and J). Close examination of blood circulation under phase-contrast imaging reveals directional blood circulation from the PrA through the ACeV via the PG vessel (PGV) which we term to specify (Figure 3K).

## BAC transgenic analysis of *vegf* expression at the developing dCP/PG interface

Since molecular mechanisms that drive the vascularization of the dCP/PG interface are unknown, we first explored where and which vegfs are expressed during the development of this interface. To visualize the expression of individual vegf isoforms at the single-cell resolution, we employed our recently generated BAC transgenic Gal4FF reporter lines that reliably recapitulate each of the endogenous gene expression detected by in situ hybridization (Parab et al., 2021; Mullapudi et al., 2019). The individual Gal4FF drivers were first crossed with Tg(UAS:EGFP-CAAX) fish, which carry a membrane-bound EGFP gene downstream of upstream activation sequence (UAS). The resultant double Tg fish are hereafter abbreviated TgBAC(vegfab:EGFP), TgBAC(vegfc:EGFP), TgBAC(vegfd:EGFP), and TgBAC(vegfaa:EGFP). All of these Tg lines were further crossed with the endothelial reporter Tg(kdrl:ras-mCherry) to visualize vegf-expressing cells and vascular endothelial cells (vECs) simultaneously. Using these Tg lines, we examined spatial relationships between vegf-expressing cells and developing vasculature.

We noted that at 26 hpf, TgBAC(vegfab:EGFP) expression is specifically localized at the junction where bilateral PrA joins (Figure 3L–L"). At 75 hpf, an increased number of TgBAC(vegfab:EGFP)⁺ cells were seen in this region (Figure 3M), which were co-labeled with an antibody for Claudin-5,

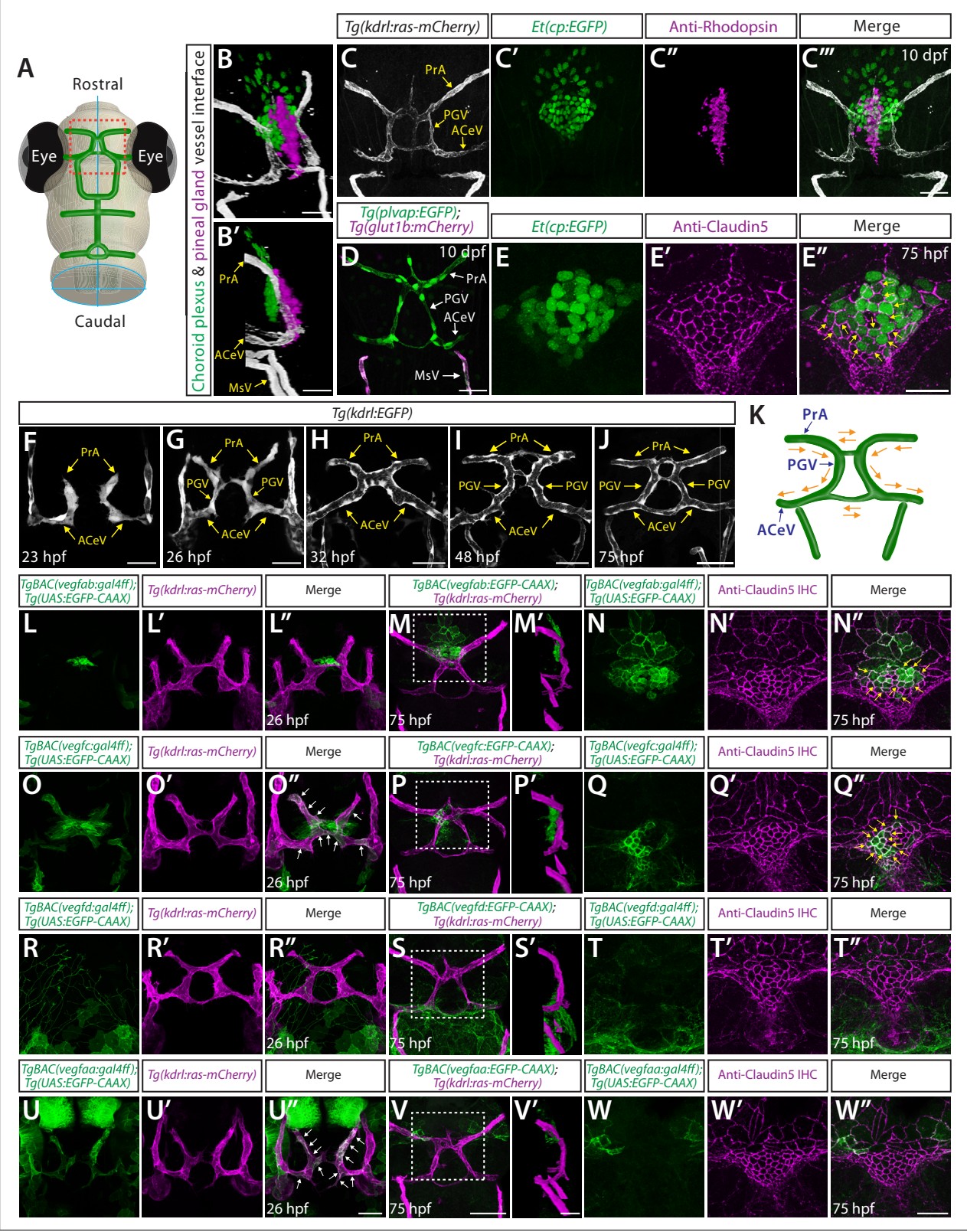

**Figure 3.** Developmental angiogenesis at the interface of the diencephalic choroid plexus (dCP) and pineal gland (PG), and *vegf* expression patterns during the vascularization of this interface. (**A**) Schematic representation of the dorsal view of the zebrafish larval head at 10 days post fertilization (dpf), indicating the location of the dCP and PG interface (red boxed area). (**B–C'''**) 3D (**B**), lateral (**B'**), and dorsal (**C–C'''**) views of the 10 dpf *Et(cp:EGFP);Tg(kdrl:ras-mCherry)* head immunostained for rhodopsin show the 3D spatial relationship between dCP epithelial cells (green),

*Figure 3 continued on next page*

*Figure 3 continued*

pineal photoreceptor cells (magenta), and blood vessels (white). (**D**) Dorsal view of 10 dpf *Tg(plvap:EGFP);Tg(glut1b:mCherry)* head shows strong *Tg(plvap:*EGFP) and undetectable *Tg(glut1b:*mCherry) expression in the prosencephalic artery (PrA), PG vessel (PGV), and anterior cerebral vein (ACeV). In contrast, the neighboring vessels mesencephalic vein (MsV) display strong *Tg(glut1b:*mCherry) expression. (**E–E″**) Dorsal views of the 75 hours post fertilization (hpf) *Et(cp:EGFP)* head immunostained for Claudin-5 show EGFP⁺ dCP epithelial cells (yellow arrows) outlined by the tight junction protein Claudin-5. (**F–J**) Dorsal views of 23 (**F**), 26 (**G**), 32 (**H**), 48 (**I**), and 75 (**J**) hpf *Tg(kdrl:EGFP)* rostral cranial vasculature show the developmental time courses of vascularization at the dCP/PG interface. (**K**) Schematic diagram of the vasculature at the dCP/PG interface shows the direction of blood flow at around 75–10 dpf. (**L–L″**) Dorsal views of a 26 hpf *TgBAC(vegfab:EGFP);Tg(kdrl:ras-mCherry)* head show *TgBAC(vegfab:*EGFP)⁺ cells at the midline where bilateral PrA connect. (**M–N″**) Dorsal (**M–N″**) and lateral (**M′**) views of a 75 hpf *TgBAC(vegfab:EGFP);Tg(kdrl:ras-mCherry)* head immunostained for Claudin-5. As compared to 26 hpf, an increased number of *TgBACvegfab:*EGFP ⁺ cells was observed at the PrA connection site and in its anterior brain regions (**M, M′**). Magnified images of the boxed area in (**M**) indicate *TgBAC(vegfab:*EGFP)⁺ and Claudin-5⁺ dCP epithelial cells (yellow arrows, **N–N″**). (**O–O″**) Dorsal views of a 26 hpf *TgBAC(vegfc:EGFP);Tg(kdrl:ras-mCherry)* head show *TgBAC(vegfc:*EGFP)⁺ vascular endothelial cells (vECs, white arrows) and separate cells at the PrA connection site. (**P–Q″**) Dorsal (**P**, **Q–Q″**) and lateral (**P′**) views of a 75 hpf *TgBAC(vegfc:EGFP);Tg(kdrl:ras-mCherry)* head immunostained for Claudin-5. *TgBAC(vegfc:*EGFP)⁺ cells were observed at the PrA connection site and in its posterior brain regions (**P, P′**). Magnified images of the boxed area in (**P**) indicate *TgBAC(vegfc:*EGFP)⁺ and Claudin-5⁺ dCP epithelial cells (yellow arrows, **Q–Q″**). (**R–R″**) Dorsal views of a 26 hpf *TgBAC(vegfd:EGFP);Tg(kdrl:ras-mCherry)* head show *TgBAC(vegfd:*EGFP)⁺ meningeal fibroblast-like cells that reside posterior to the dCP/PG interface. *TgBAC(vegfd:*EGFP)⁺ axonal projections were also visualized. (**S–T″**) Dorsal (**S, T–T″**) and lateral (**S′**) views of a 75 hpf *TgBAC(vegfd:EGFP);Tg(kdrl:ras-mCherry)* head immunostained for Claudin-5. *TgBAC(vegfd:*EGFP)⁺ cells were observed in meningeal fibroblast-like cells that reside posterior to the PGV/ACeV (**S, S′**). Magnified images of the boxed area in (**S**) show no obvious *TgBAC(vegfd:*EGFP) expression in dCP epithelial cells (**T–T″**). (**U–U″**) Dorsal views of a 26 hpf *TgBAC(vegfaa:EGFP);Tg(kdrl:ras-mCherry)* head show *TgBAC(vegfaa:*EGFP)⁺ vECs (white arrows). (**V–W″**) Dorsal (**V, W–W″**) and lateral (**V′**) views of a 75 hpf *TgBAC(vegfaa:EGFP);Tg(kdrl:ras-mCherry)* head immunostained for Claudin-5. Sparse *TgBAC(vegfaa:*EGFP)⁺ cells were observed at the lateral periphery of the dCP (**W–W′**). Magnified images of the boxed area in (**V**) show *TgBAC(vegfaa:*EGFP)⁺ and Claudin-5⁺ dCP epithelial cells at the periphery (**W-W′**). Scale bars: 30 μm in (**B**), (**B′**), (**D**), in (**C″′**) for (**C–C″′**), in (**E″**) for (**E–E″**), in (**V′**) for (**M′**), (**P′**), (**S′**); 50 μm in (**F–J**), in (**U″**) for (**L–L″**), (**O–O″**), (**R–R″**), (**U–U″**), in (**V**) for (**M**), (**P**), (**S**), in (**W″**) for (**N–N″**), (**Q–Q″**), (**T–T″**), (**W–W″**).

a marker for dCP epithelial cells (*Figure 3E–E″ and N–N″*). On the other hand, we observed the broader *TgBAC(vegfc:*EGFP) expression in two cell types around this region at 26 hpf (*Figure 3O–O″*): (1) developing vECs that constitute the ACeV, PGV, and PrA and (2) cells at the junction where bilateral PrA joins. By 75 hpf, *TgBAC(vegfc:*EGFP) expression in vECs disappeared (*Figure 3P*), although *TgBAC(vegfc:*EGFP)⁺ cells marked by Claudin-5 immunoreactivity were still observed (*Figure 3Q–Q″*). These *TgBAC(vegfc:*EGFP)⁺ cells lie at the PrA junction slightly posterior to where *TgBAC(vegfab:*EGFP) expression was detected (*Figure 3M′ and P′*). In contrast, *TgBAC(vegfd:*EGFP) expression was not observed at the PrA junction or in dCP epithelial cells at 26 and 75 hpf (*Figure 3R–T″*). *TgBAC(vegfaa:*EGFP) expression was found in developing vECs that constitute the ACeV, PGV, and PrA at 26 hpf (*Figure 3U–U″*). This endothelial *TgBAC(vegfaa:*EGFP) expression disappeared by 75 hpf (*Figure 3V and V′*), and only sparsely labeled cells were observed in the region close to the dCP (*Figure 3W–W″*).

To summarize, *TgBAC(vegfc:EGFP)* and *TgBAC(vegfaa:EGFP)* expression was observed similarly in extending vECs at 26 hpf, which disappeared by 75 hpf. *TgBAC(vegfc:EGFP)* and *TgBAC(vegfab:EGFP)* expression at 26 hpf appears to mark cells that subsequently differentiate into dCP epithelial cells by 75 hpf. These overlapping expression patterns imply potential functional redundancy among multiple Vegf ligands in regulating vascularization at the dCP/PG interface.

## Anatomically separate CPs require a distinct set of Vegf ligands for vascularization

We previously demonstrated that multiple Vegf ligands (Vegfab, Vegfc, and Vegfd) are redundantly required for the angiogenesis leading to fenestrated mCP vasculature (*Parab et al., 2021*).

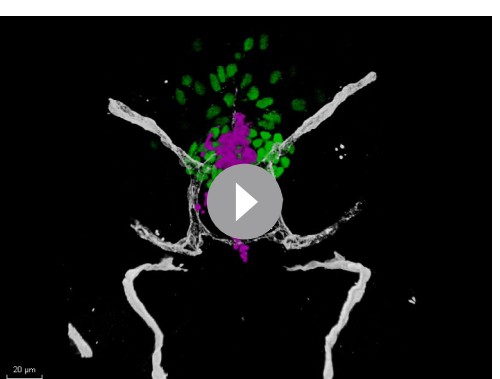

**Video 1.** Fenestrated vasculature at the pineal gland (PG)/diencephalic choroid plexus (dCP) interface. Ten days post fertilization (dpf) *Et(cp:EGFP);Tg(kdrl:ras-mCherry)* head immunostained for rhodopsin shows the 3D spatial relationship between dCP epithelial cells (green), pineal photoreceptor cells (magenta), and blood vessels (white).

https://elifesciences.org/articles/86066/figures#video1

To determine if this molecular combination is a universal inducer of fenestrated vessel formation across the brain, we first asked whether the same set of Vegf ligands is required for dCP/PG vascularization. To this aim, we analyzed *vegfab*[bns92], *vegfc*[hu6410], and *vegfd*[bns257] mutants that carried the endothelial *Tg(kdrl:EGFP)* reporter individually and in all possible combinations. We chose to analyze these mutants at 10 dpf – a stage when most of the major blood vessels are formed (*Isogai et al., 2001*) and that is a week after vasculature at the dCP/PG interface is normally established. Conducting a phenotypic analysis at this stage allowed us to eliminate the possibility of developmental delays in mutants.

At 10 dpf, none of the *vegfab*[bns92], *vegfc*[hu6410], and *vegfd*[bns257] single mutants displayed a drastic difference compared to WT, except a mild, yet significant, defect in PrA formation in *vegfc*[hu6410] mutants (*Figure 4A–D and I*). The combined loss of the three genes led to significantly exacerbated phenotypes in fenestrated vessel formation at the dCP/PG interface (*Figure 4H*). The most pronounced defect observed was on PrA formation, although this phenotype is partially penetrant (*Figure 4I*). One-third of the triple mutants lacked bilateral PrA, and 40% triple mutants exhibited only unilateral PrA. ACeV formation deficit is milder in triple mutants, with approximately 27% of them exhibiting unilateral ACeV. Statistical analysis of these genetic data shows that *vegfab* and *vegfc* genetically interact in PrA formation with little effect on PGV and ACeV formation. In contrast, *vegfd* and *vegfc* genetically interact in regulating PGV and ACeV, but not PrA, formation. While these results indicate the requirements of all the three Vegf ligands for vascularization of this interface, the phenotypes observed in triple mutants are much milder than those observed in the mCP where we found a near-complete penetrance and loss of strong *Tg(plvap:*EGFP)[+] vasculature (*Parab et al., 2021*). These data imply differential molecular requirements for fenestrated capillary formation across the CPs.

To seek additional angiogenic factors critical for vascularization of the dCP/PG interface, we analyzed mutants that lack another Vegfa paralog, Vegfaa. Since *vegfaa*[bns1] mutants were previously shown to die at approximately 5 dpf due to the severe early embryonic vascular defects (*Rossi et al., 2016*), we analyzed this mutant at 72 hpf. *vegfaa*[bns1] single mutants did not show a defect in vascularization at the dCP/PG interface (*Figure 4K and R*). However, the mutants displayed a severe defect in the formation of the neighboring bilateral MsV, which exhibits robust *Tg(glut1b:*mCherry) expression and immunoreactivity of several BBB markers (*Figures 1C", E and 3D*; *Figure 1—figure supplement 1A–D*; *Parab et al., 2021*). When we deleted *vegfaa* and *vegfc* simultaneously, PrA and PGV formation defects were substantially enhanced (*Figure 4P*) and were further exacerbated by the additional deletion of *vegfab* (*Figure 4Q*). Statistical analysis supports a strong genetic interaction between *vegfc* and *vegfaa* in the development of the PrA and PGV, but not of the ACeV. The combined loss of *vegfab* and *vegfc* induced striking defects specifically in PrA formation at 72 hpf similar to what was observed at 10 dpf (*Figure 4O*). Intriguingly, the combined loss of *vegfaa* and *vegfab* did not cause a defect in vascularization at this interface (*Figure 4N and R*).

Collectively, these mutant analyses uncover remarkably heterogeneous endothelial requirements for angiogenesis within this local brain environment. Specifically, the results reveal Vegfc's genetic interactions with *vegfab*, *vegfd*, or *vegfaa* in regulating fenestrated vessel formation at the dCP/PG interface, placing Vegfc as a central angiogenic regulator of this process. This angiogenic requirement is different from that needed for mCP vascularization which involves a combination of Vegfab, Vegfc, and Vegfd in the way that Vegfab acts as a central angiogenic factor. This distinction may be due to brain regional differences in CP molecular signatures, including secretome, as indicated by the recent transcriptomic studies of CPs dissected from different ventricles (*Lun et al., 2015*; *Dani et al., 2021*).

## Endothelial cell-autonomous and cell non-autonomous requirements of Vegfc for vascularization of the dCP/PG interface

Our expression and genetic data indicate that *vegfc* is expressed in both developing vECs and dCP epithelial cells during dCP/PG vascularization and that Vegfc functionally interacts with other Vegfs in controlling this process. Indeed, we observed *vegfab* expression in developing dCP epithelial cells and *vegfaa* expression in extending vECs during dCP/PG vascularization. These observations led us to hypothesize that Vegfc's angiogenic activity involves both endothelial cell-autonomous and cell non-autonomous actions that require the discrete Vegfa paralogs.

To test this hypothesis, we employed a separate *vegfc* mutation, *vegfc*[um18], which was previously isolated from a forward genetic screen (*Villefranc et al., 2013*). This particular mutation was shown to generate a prematurely truncated Vegfc protein that lacks efficient secretory and paracrine activity but

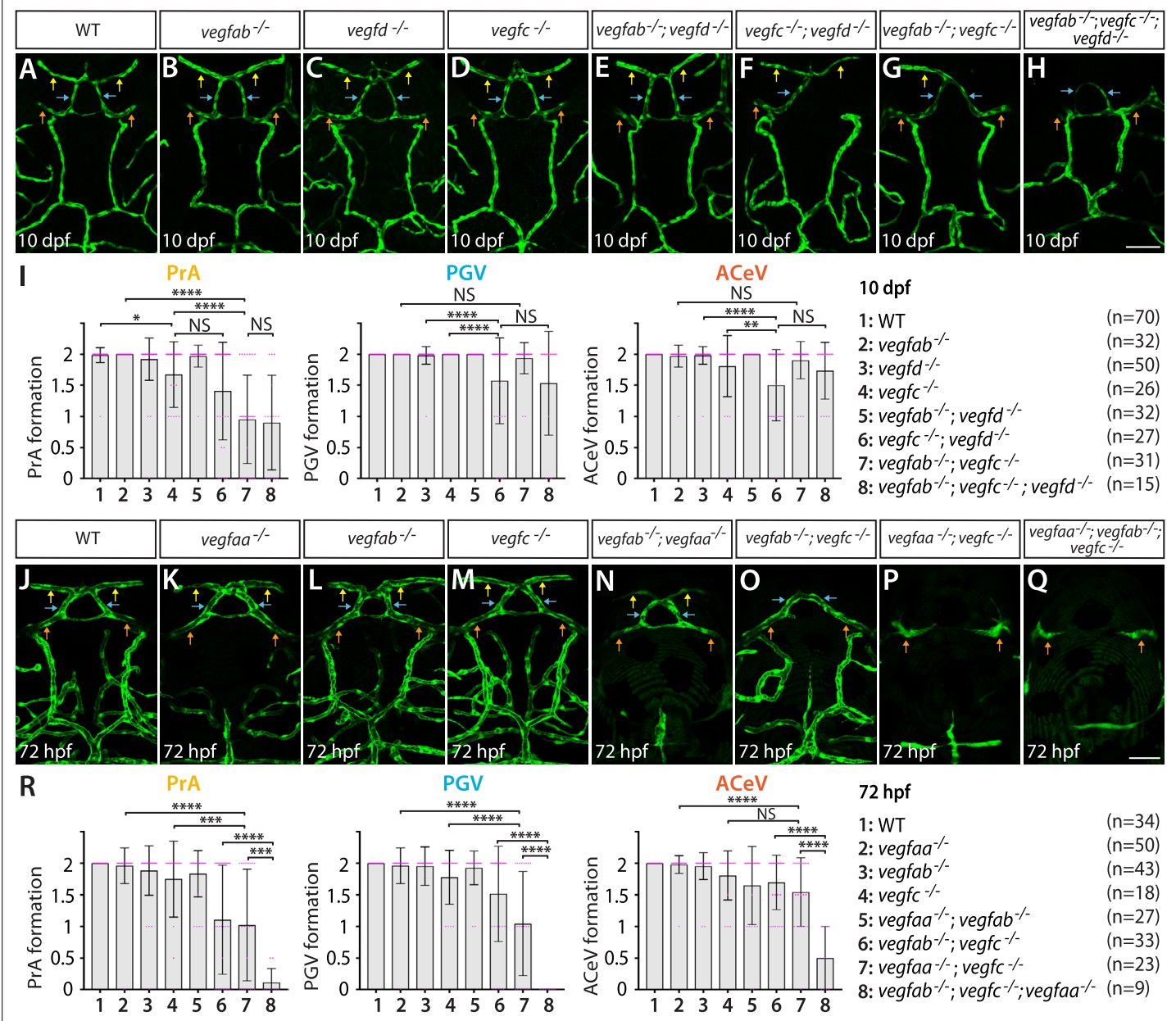

**Figure 4.** Heterogeneous endothelial requirements for Vegfs-dependent angiogenesis at the diencephalic choroid plexus (dCP)/pineal gland (PG) interface. (**A–H**) Dorsal views of 10 days post fertilization (dpf) wild-type (WT) (**A**), vegfab⁻/⁻ (**B**), vegfd⁻/⁻ (**C**), vegfc⁻/⁻ (**D**), vegfab⁻/⁻;vegfd⁻/⁻ (**E**), vegfc⁻/⁻;vegfd⁻/⁻ (**F**), vegfab⁻/⁻;vegfc⁻/⁻ (**G**), and vegfab⁻/⁻;vegfc⁻/⁻;vegfd⁻/⁻ (**H**) cranial vasculature visualized by *Tg(kdrl*:EGFP) expression. Yellow arrows point to the prosencephalic artery (PrA), blue arrows to the PG vessel (PGV), and orange arrows to the anterior cerebral vein (ACeV). A majority of vegfab⁻/⁻;vegfc⁻/⁻ (**G**) and vegfab⁻/⁻;vegfc⁻/⁻;vegfd⁻/⁻ (**H**) larvae lacked the PrA at either or both sides. vegfc⁻/⁻;vegfd⁻/⁻ (**F**) and vegfab⁻/⁻;vegfc⁻/⁻;vegfd⁻/⁻ (**H**), but not vegfab⁻/⁻;vegfc⁻/⁻ (**G**), larvae displayed partially penetrant defects in PGV and/or ACeV formation. (**I**) Quantification of PrA, PGV, and ACeV formation at 10 dpf (the number of animals examined per genotype is listed in the panel). Statistical data support genetic interactions between vegfab and vegfc in PrA formation and between vegfd and vegfc in PGV and ACeV formation. No significant contributions of vegfd or vegfab to the formation of the PrA or PGV/ACeV, respectively, were noted. (**J–Q**) Dorsal views of 72 hours post fertilization (hpf) WT (**J**), vegfaa⁻/⁻ (**K**), vegfab⁻/⁻ (**L**), vegfc⁻/⁻ (**M**), vegfab⁻/⁻;vegfaa⁻/⁻ (**N**), vegfab⁻/⁻;vegfc⁻/⁻ (**O**), vegfaa⁻/⁻;vegfc⁻/⁻ (**P**), and vegfaa⁻/⁻;vegfab⁻/⁻;vegfc⁻/⁻ (**Q**) cranial vasculature visualized by *Tg(kdrl*:EGFP) expression. Yellow arrows point to the PrA, blue arrows to the PGV, and orange arrows to the ACeV. vegfab⁻/⁻;vegfc⁻/⁻, but not their respective single mutants, exhibited pronounced PrA formation deficits. vegfaa⁻/⁻ and vegfab⁻/⁻;vegfaa⁻/⁻ displayed severe defects in mesencephalic vein (MsV) formation without a deficit in PrA, PGV, or ACeV development. vegfaa⁻/⁻;vegfc⁻/⁻ and vegfaa⁻/⁻;vegfab⁻/⁻;vegfc⁻/⁻ larvae exhibited a severe loss of the PrA and PGV. (**R**) Quantification of PrA, PGV, and ACeV formation at 72 hpf (the number of animals examined per genotype is listed in the panel). Statistical data support genetic interactions between vegfab and vegfc in PrA formation and between vegfaa and vegfc in PrA and PGV formation. Furthermore, significant genetic interactions among these three genes were detected in vascularization at this interface. In panels (**I and R**), each data point shown in magenta

*Figure 4 continued on next page*

Figure 4 continued

represents individual animal's vessel formation score, and values represent means ± SD (*, **, ***, and **** indicate p<0.05, p<0.01, p<0.001, and p<0.0001, respectively, by one-way analysis of variance [ANOVA] followed by Tukey's HSD test). Scale bars: 50 μm in (H) for (A–H) and in (Q) for (J–Q).

retains the ability to activate its Flt4 receptor (in another words, cell-autonomous activity of Vegfc is retained) (*Villefranc et al., 2013*). Homozygous embryos carrying this mutation alone did not exhibit a defect in dCP/PG vascularization at 72 hpf (*Figure 5A and B*). However, we noted that the *vegfc*[um18] mutant allele displayed a genetic interaction with *vegfab*, leading to a partially penetrant, yet significant, defect in PrA formation (*Figure 5A–F*). The extent of this PrA defect is milder than that was noted in double mutants of *vegfab;vegfc*[hu6410] larvae analyzed at the same stage (*Figure 5I*). These results suggest that reduced paracrine activity of Vegfc causes a milder defect than that observed in fish that lack both paracrine and autocrine Vegfc activities, indicating that endothelial cell-autonomous and cell non-autonomous Vegfc activities are both necessary for PrA formation.

We next analyzed double mutants of *vegfaa;vegfc*[um18] larvae to gain additional insight into Vegfc angiogenic action. We found that *vegfaa;vegfc*[um18] double mutant larvae exhibited a similar extent of the defect observed in *vegfaa;vegfc*[hu6410] double mutant larvae examined at the same stage (*Figure 5G–I*), suggesting that a loss of Vegfc paracrine activity has a profound impact on PrA formation in *vegfaa* mutant background. We noted that the formation of neighboring blood vessels, MsV, which displays medium levels of both BBB and fenestrated marker expression, was severely abrogated in the absence of Vegfaa alone (*Figures 5G and 4K*) and further exacerbated in combination with Vegfab or Vegfc deletion (*Figure 5H and J*). These results show that Vegfaa plays a major role in regulating MsV formation, providing additional evidence for heterogeneous endothelial requirements for angiogenesis within this local brain region. Various vascular phenotypes observed in *vegf* mutants are summarized in *Figure 5K*.

## *vegf* BAC transgenic expression analysis in ventral brain regions around the NH and OVLT

The NH constitutes the hypophysis (pituitary gland) and forms a neurovascular interface where neuropeptides are secreted into blood circulation via fenestrated capillaries, which regulate reproduction, fluid balance, and blood pressure (*Biran et al., 2018*; *Grinevich and Dobolyi, 2021*; *Pearson and Placzek, 2013*). Prior work implies that the OVLT, a sensory CVO crucial for fluid homeostasis via osmotic sensing and regulation (*Noda and Sakuta, 2013*; *McKinley and Johnson, 2004*; *Zimmerman et al., 2017*), resides dorsally to the rostral part of hypophyseal artery (HyA) (*García-Lecea et al., 2017*). To date, angiogenic cues that drive fenestrated capillary development in the NH and OVLT remain unclear.

A previous study characterized the developmental process of capillary formation at the hypophysis (*Gutnick et al., 2011*). HyA formation begins by sprouting from the cranial division of the internal carotid artery toward the midline, which extends rostrally until it connects with the palatocerebral arteries (PLA) at around 60 hpf (*Gutnick et al., 2011*). Bilateral HyA fuse with hypophyseal veins (HyV) to form the vascular loop at the hypophysis (Hy loop) by 72 hpf (*Gutnick et al., 2011*). Ultrastructural analysis of the hypophysis in both larval and adult zebrafish revealed capillary fenestrations in this organ (*Gordon et al., 2019*; *Anbalagan et al., 2018*). In support of this observation, we noted strong *Tg(plvap:*EGFP) expression in vECs that comprise the Hy loop and HyA at 120 hpf (*Figure 2B*), although much fainter *Tg(plvap:*EGFP) expression was seen in HyV.

We examined expression patterns of *vegfaa*, *vegfab*, *vegfc*, and *vegfd* using BAC transgenic reporters at several developmental stages and observed distinct and partially overlapping expression patterns in the ventral diencephalon. Prominent *vegfaa* expression was detected in cells that reside in close proximity to the Hy loop (*Figure 6A and E*). Strong *vegfab* expression was found in cells that lie rostrodorsally to the anterior part of the Hy loop (*Figure 6B and F*) and also in cells that reside at the junction where HyA extends to join the PLA (*Figure 6G*). Notable *vegfc* expression was observed in several different cell types that reside closely to the rostral portion of the HyA (*Figure 6C and H*). Expression of *vegfd* BAC reporter was not detectable in this brain region at the developmental stages we examined (*Figure 6D*). Moreover, *vegfaa*, *vegfab*, and *vegfc* BAC reporter expression was not observed in vECs that constitute HyA, HyV, or Hy loop.

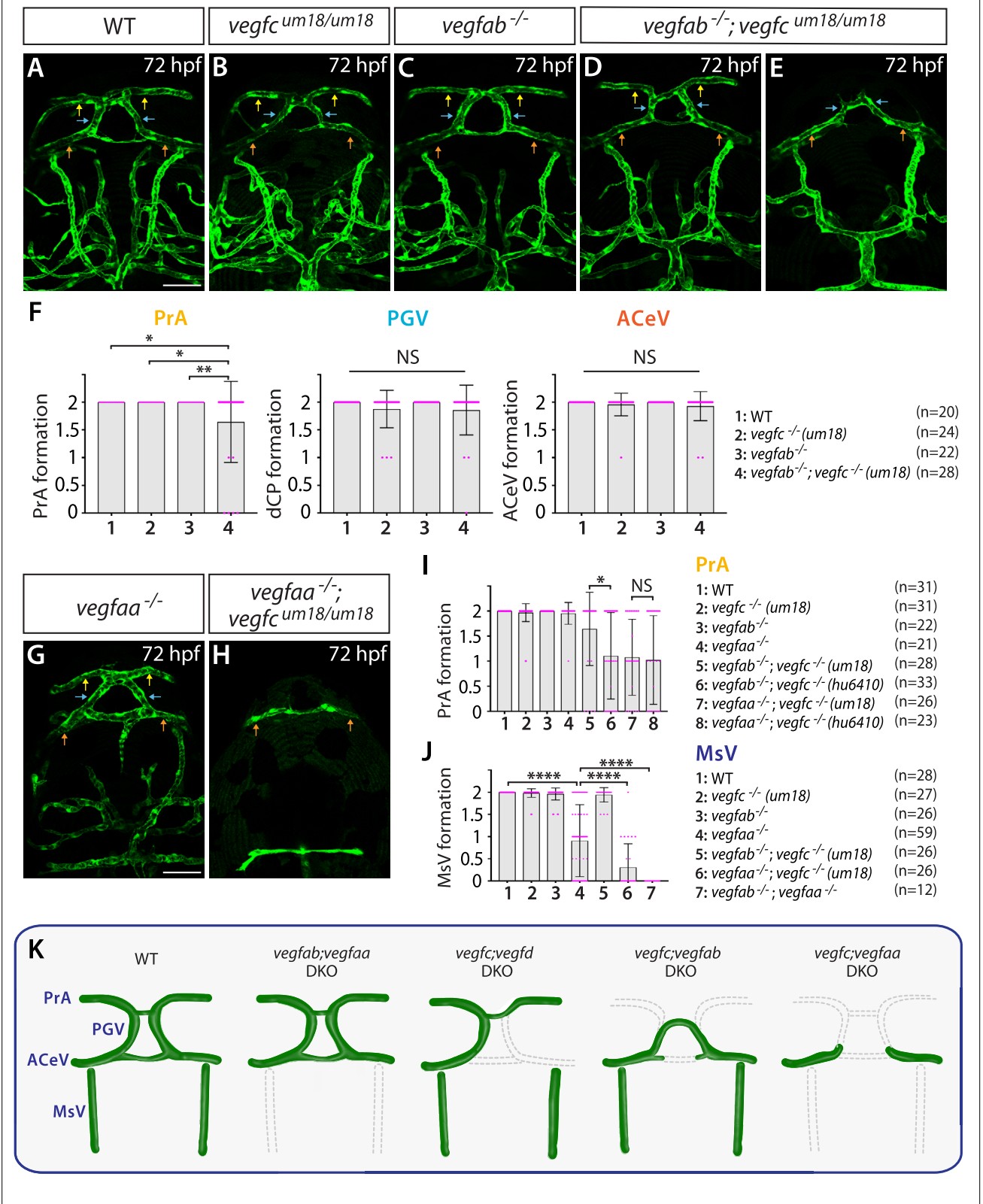

**Figure 5.** Endothelial cell-autonomous and cell non-autonomous requirements of Vegfc for vascularization of the diencephalic choroid plexus (dCP)/pineal gland (PG) interface. (A–E) Dorsal views of 72 hours post fertilization (hpf) wild-type (WT) (A), *vegfc*[um18/um18] (B), *vegfab*[-/-] (C), and *vegfab*[-/-];*vegfc*[um18/um18] (D, E) cranial vasculature visualized by *Tg(kdrl*:EGFP) expression. Yellow arrows point to the prosencephalic artery (PrA), blue arrows to the PG vessel (PGV), and orange arrows to the anterior cerebral vein (ACeV). Although none of *vegfc*[um18/um18] and *vegfab*[-/-] fish exhibited a defect in PrA

*Figure 5 continued*

formation (**B, C**), approximately 21% of *vegfab^-/-^;vegfc^um18/um18^* larvae lacked the PrA at either or both sides (**E**). (**F**) Quantification of PrA, PGV, and ACeV formation at 72 hpf (the number of animals examined per genotype is listed in the panel). Specific defect was observed in PrA formation in *vegfab^-/-^;vegfc^um18/um18^* larvae compared to other three genotypes. (**G, H**) Dorsal views of 72 hpf *vegfaa^-/-^* (**G**) and *vegfaa^-/-^;vegfc^um18/um18^* (**H**) cranial vasculature visualized by *Tg(kdrl:EGFP)* expression. Yellow arrows point to the PrA, blue arrows to the PGV, and orange arrows to the ACeV. Although *vegfaa^-/-^* or *vegfc^um18/um18^* larvae fully formed vasculature at the dCP/PG interface, most of *vegfaa^-/-^;vegfc^um18/um18^* larvae failed to form the PrA and PGV at either or both sides. (**I**) Quantification of PrA formation at 72 hpf (the number of animals examined per genotype is listed in the panel). The quantitative results of several genotypes were presented again or integrated in this graph for comparison purposes. Previously presented results are the *vegfc^-/-^* (*hu6410* allele) data from **Figure 4R**, and the data in (**F**) were either re-presented or combined with the quantitative results shown in **Figure 5G and H**. Paracrine activity-deficient *vegfc^um18/um18^* larvae in the *vegfab^-/-^* background displayed a significantly milder defect in PrA formation than that observed in *vegfab^-/-^;vegfc^-/-^* (*hu6410* allele) animals that lack both endothelial cell-autonomous and cell non-autonomous Vegfc function. (**J**) Quantification of mesencephalic vein (MsV) formation at 72 hpf (the number of animals examined per genotype is listed in the panel). Severe defects in MsV formation in *vegfaa^-/-^* larvae were further exacerbated by genetic deletions of *vegfc* (*um18* allele) or *vegfab*. (**K**) Schematic representations of the severe vascular phenotypes observed in 72 hpf various *vegf* mutants at the dCP/PG interface. Genetic results indicate highly heterogeneous molecular requirements for angiogenesis around the dCP/PG interface. In panels (**F**), (**I**), and (**J**), each data point shown in magenta represents individual animal's vessel formation score, and values represent means ± SD (*, **, and **** indicate p<0.05, p<0.01, and p<0.0001, respectively, by one-way analysis of variance [ANOVA] followed by Tukey's HSD test). Scale bars: 50 μm in (**A**) for (**A–E**) and in (**G**) for (**G–H**).

To determine cell types that express these *vegf* genes, we carried out the following experiments. First, we performed in situ hybridization on these BAC transgenic reporters using cell type-specific markers. We used a *cyp26b1* probe for marking the astroglial pituicyte and a *prop1* probe for progenitors, as used in recent studies (*Anbalagan et al., 2018*; *Chen et al., 2022*; *Fabian et al., 2020*). Although we did not observe any overlap between *prop1^+^* and *vegfaa* or *vegfab* BAC reporter^+^ cells at the hypophysis (**Figure 6K–P"**), we found significant co-localizations between *cyp26b1^+^* pituicytes and these reporter^+^ cells (**Figure 6I–J"'**; **Figure 6—figure supplement 1**), suggesting that pituicytes are a source of these Vegf ligands. This result is consistent with the recently published bulk- and scRNA-seq data that show *vegfaa* and *vegfab* expression in pituicytes (*Anbalagan et al., 2018*; *Chen et al., 2022*; *Fabian et al., 2020*). Additionally, we injected a plasmid at the one-cell stage, which drives mScarlet expression under the *pomc* promoter that was previously shown to induce gene expression in pituitary corticotrophs (*Liu et al., 2003*). We observed no co-localization between *pomc^+^* and *vegfaa^+^* or *vegfab^+^* cells at the hypophysis (**Figure 6—figure supplement 1Q-S"**), indicating that pituitary corticotrophs are not a cell type that expresses the *vegfa* genes. A series of these co-localization experiments suggest that pituicytes are the major cell type expressing *vegfaa* and *vegfab* at the hypophysis. Importantly, scRNA-seq data of the adult NH in mice showed pituicyte-specific expression of *Vegfa*, indicating interspecies conservation of *vegfa* expression in pituicytes (*Chen et al., 2020*).

## Distinct endothelial requirements for Vegfs-dependent angiogenesis between the dCP/PG and NH/OVLT brain regions

Next, we analyzed *vegfaa*, *vegfab*, *vegfc^hu6410^*, and *vegfd* mutants individually and in various combinations to determine the requirements of Vegf ligands for vascularization of the NH/OVLT brain regions. We noted that none of these individual *vegf* mutants displayed an obvious defect in the formation of the HyA, Hy loop, or HyV (**Figure 7A–D and I**). Although the combined deletion of *vegfab*, *vegfc*, and *vegfd* caused no defect in the HyV and Hy loop, we found some triple mutants (7 out of 18, approximately 39%) exhibiting either absence or partial formation of the HyA (**Figure 7H and I**). A comparable phenotype was observed even in *vegfab;vegfc* double mutants (7 out of 22, approximately 32%) (**Figure 7G and I**), suggesting that this HyA formation defect results from a genetic interaction between *vegfab* and *vegfc*. Contribution of Vegfd to this vascularization process is likely absent or minor since no genetic interaction between *vegfd* and *vegfab* or *vegfc* was noted (**Figure 7E, F and I**). These genetic data are in line with undetectable expression of the *vegfd* BAC reporter.

Since *vegfab;vegfc;vegfd* triple or *vegfab;vegfc* double mutants displayed only a partially penetrant defect in HyA formation, we next investigated the role of Vegfaa in this process. Interestingly, we found that *vegfaa* single mutants exhibited either absent or stalled HyA phenotypes (38 out of 48, approximately 79%) (**Figure 7M**), similar to *vegfab^-/-^;vegfc^-/-^* larvae, but to a greater extent. However, this HyA formation defect in *vegfaa* single mutants is still partially penetrant (**Figure 7Q**), allowing us to test combined deletions of *vegfaa*, *vegfab*, and/or *vegfc^hu6410^*. We observed that the severity of HyA stalling phenotype in *vegfaa^-/-^* larvae was substantially exacerbated by the simultaneous deletion

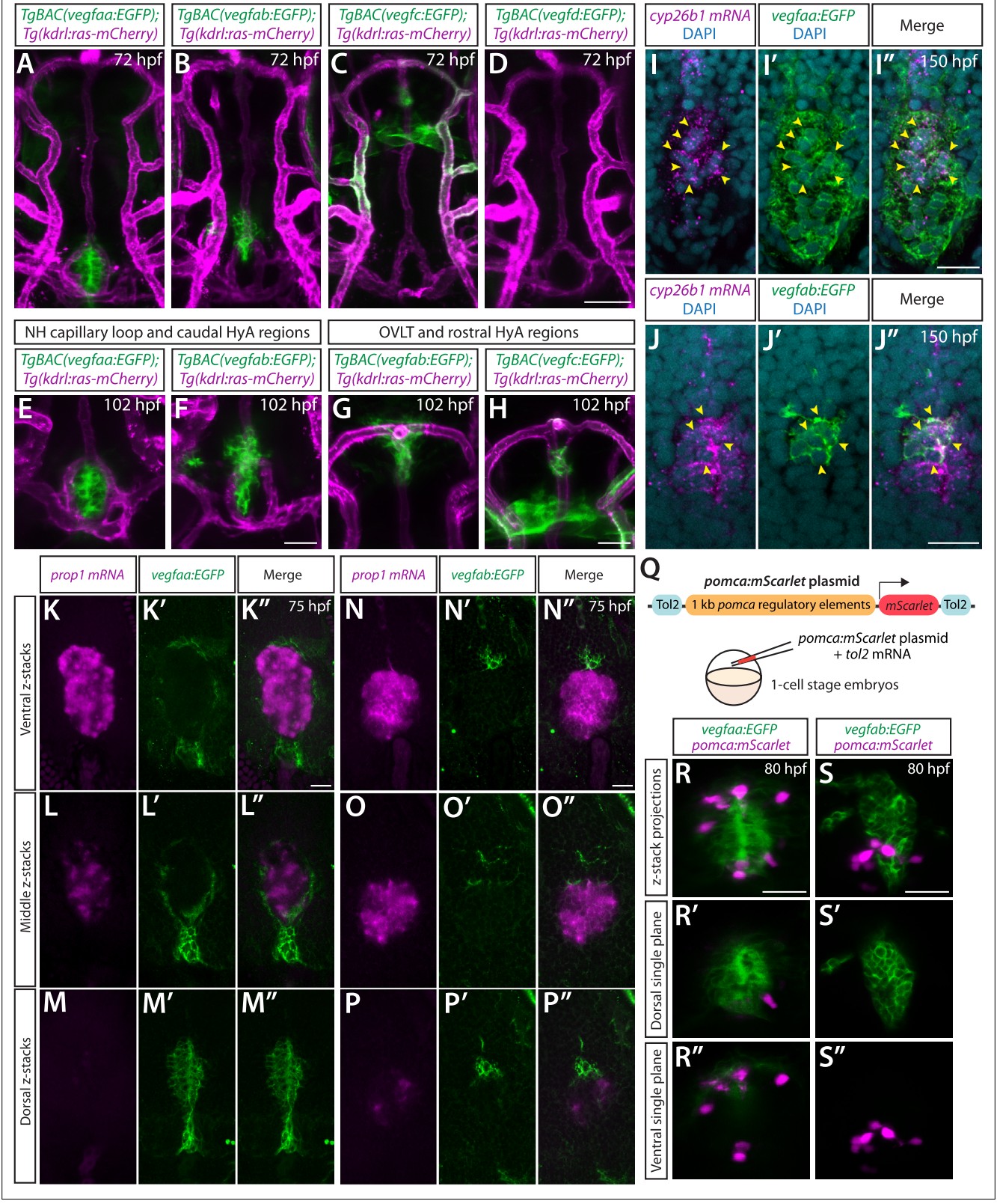

**Figure 6.** BAC transgenic analysis of *vegf* expression in the ventral brain around the neurohypophysis (NH)/organum vasculosum of the lamina terminalis (OVLT). (**A–D**) Dorsal views of 72 hours post fertilization (hpf) *TgBAC(vegfaa:EGFP)* (**A**), *TgBAC(vegfab:EGFP)* (**B**), *TgBAC(vegfc:EGFP)* (**C**), and *TgBAC(vegfd:EGFP)* (**D**) ventral brain of the larvae carrying the *Tg(kdrl:ras-mCherry)* transgene. Prominent *TgBAC(vegfaa:*EGFP) and *TgBAC(vegfab:*EGFP) expression was observed in cells that reside in close proximity to Hy loop at the NH (**A, B**). Notable *TgBAC(vegfc:*EGFP)

*Figure 6 continued on next page*

*Figure 6 continued*

expression was detected in the rostral portion of the hypophyseal artery (HyA) around the OVLT (**C**). *TgBAC(vegfd:*EGFP) expression was not detectable at this stage (**D**). (**E–H**) Magnified images of 102 hpf *TgBAC(vegfaa:*EGFP) (**E**), *TgBAC(vegfab:*EGFP) (**F, G**), and *TgBAC(vegfc:*EGFP) (**H**) ventral brain of the larvae carrying the *Tg(kdrl:ras-mCherry)* transgene. While *TgBAC(vegfaa:*EGFP)⁺ cells reside slightly dorsal to the Hy loop (**E**), many *TgBAC(vegfab:*EGFP)⁺ cells were located further rostrodorsally likely in the hypothalamus (**F**). In the rostral part of the HyA around the HyA-palatocerebral arteries (PLA) junction and OVLT, *TgBAC(vegfab:*EGFP) and *TgBAC(vegfc:*EGFP) expression was detected in peri-vascular cells (**G, H**). Additionally, strong *TgBAC(vegfc:*EGFP) signals were observed around the OVLT. (**I–I″**) Single confocal z-plane images of 150 hpf *TgBAC(vegfaa:*EGFP) ventral brain following in situ hybridization of *cyp26b1*, showing overlapping signals (yellow arrowheads) between the EGFP⁺ cells and *cyp26b1*⁺ pituicyte (n=10). (**J–J″**) Single confocal z-plane images of 150 hpf *TgBAC(vegfab:*EGFP) larval ventral brain following in situ hybridization of *cyp26b1*, showing overlapping signals (yellow arrowheads) between the EGFP⁺ cells and *cyp26b1*⁺ pituicyte (n=11). (**K–M″**) Serial confocal z-stacks of ventral (**K–K″**), middle (**L–L″**), and dorsal (**M–M″**) images showing no overlapping signals between *TgBAC(vegfaa:*EGFP)⁺ and *prop1*⁺ by in situ hybridization in 75 hpf larvae (n=8). (**N–P″**) Serial confocal z-stacks of ventral (**N–N″**), middle (**O–O″**), and dorsal (**P–P″**) images showing no overlapping signals between *TgBAC(vegfab:*EGFP)⁺ and *prop1*⁺ by in situ hybridization in 75 hpf larvae (n=10). (**Q**) Schematic of the *pomca:mScarlet* construct used for injection experiments (**R–S″**). (**R–S″**) Magnified dorsal views of 80 hpf *TgBAC(vegfaa:*EGFP) (**R–R″**) and *TgBAC(vegfab:*EGFP) (**S–S″**) NH of larvae injected with the *pomca:mScarlet* construct at the one-cell stage. Confocal z-stack maximum projection (**R, S**) and their dorsal (**R′, S′**) and ventral (**R″, S″**) single z-plane images showed no overlapping signals between EGFP⁺ cells and *pomca:*mScarlet⁺ pituitary corticotrophs. Scale bars: 50 μm in (**D**) for (**A–D**); 25 μm in (**F**) for (**E–F**), in (**H**) for (**G–H**), in (**R**) for (**R–R″**), in (**S**) for (**S–S″**); and 15 μm in (**I″**) for (**I–I″**), in (**J″**) for (**J–J″**), in (**K″**) for (**K–M″**), in (**N″**) for (**N–P″**).

The online version of this article includes the following figure supplement(s) for figure 6:

**Figure supplement 1.** Co-localization of the pituicyte marker *cyp26b1* and *vegfaa* or *vegfab* BAC transgenic reporter expression at early larval stages.

of *vegfab* (*Figure 7L and O*), but not of *vegfc* (*Figure 7K and N*). These results show that there is a genetic interaction between *vegfaa* and *vegfab*, but not between *vegfaa* and *vegfc* (*Figure 7Q*). Intriguingly, we noted that the phenotypes observed in *vegfab⁻/⁻;vegfaa⁻/⁻* larvae were restricted to fenestrated Hy loop and the HyA with no defect in HyV formation (*Figure 7O–Q*). This finding is in stark contrast to the dCP/PG interface where BBB markers⁺ MsV was selectively impaired without any noticeable defect in fenestrated vessel formation in *vegfab⁻/⁻;vegfaa⁻/⁻* larvae (*Figure 4N*). Thus, there is a clear distinction in fenestrated endothelial requirements for Vegfs-dependent angiogenesis between the NH/OVLT and dCP/PG regions. The selective loss of fenestrated Hy loop and the HyA in *vegfab⁻/⁻;vegfaa⁻/⁻* larvae suggest that molecularly distinct Hy loop/HyA and HyV depend on discrete angiogenic mechanisms (*Figure 7P*).

## Paracrine angiogenic activity of Vegfc is crucial for HyA formation

To gain mechanistic insight into Vegfc function in regulating HyA formation, we examined paracrine activity-deficient *vegfc^um18* mutants in combination with *vegfab* mutants. We observed that *vegfc^um18* single mutants did not show any defect in HyA formation (*Figure 8A and B*). However, approximately 22% of *vegfab;vegfc^um18* double mutants (8 out of 36) displayed stalled HyA phenotypes (*Figure 8C*) similar to what was observed in *vegfab;vegfc^hu6410* double mutant larvae examined at the same stage (*Figure 8D*). These genetic results suggest that paracrine actions of Vegfc are crucial for HyA formation, which is in agreement with prominent *vegfc* BAC reporter expression in non-vECs that lie in close proximity to the rostral part of the HyA and the OVLT (*Figure 6C and H*).

Since *vegfaa;vegfab* double mutants display gross vascular defects during embryogenesis, we next tested whether temporal inhibition of Vegfa signaling could recapitulate the HyA and Hy loop defects observed in these double mutants. To this aim, we employed the previously established *Tg(hsp70l:sflt1)* and *Tg(hsp70l:sflt4)* lines in which the Vegfa and Vegfc ligand trap, sFlt1 and sFlt4, respectively, is overexpressed upon a heatshock treatment (*Matsuoka et al., 2016*; *Matsuoka et al., 2017*). To examine their effects on vascularization of the NH/OVLT, we subjected embryos to a heatshock at 34, 44, and 54 hpf, and analyzed them at 72 or 144 hpf (*Figure 8E*). We observed that heatshock-induced overexpression of sFlt1 led to stalled HyA formation toward the PLA, accompanied by a drastic reduction in the number of vECs that comprise the HyA (*Figure 8F–H, J and K*). Milder, yet pronounced, reduction of vEC numbers in the Hy loop was detected in heatshock-treated *Tg(hsp70l:sflt1)* larvae, but only small differences were recognized in the HyV of these larvae. In contrast, sFlt4 overexpression led to no difference in the number of vECs that comprise the Hy loop and HyV compared to controls, while we observed a significant reduction in HyA vEC numbers at 144 hpf (*Figure 8I–K*). These sFlt4 overexpression results indicate Vegfc's selective contribution to vEC development in HyA, which is in line with the specific defect in HyA formation observed in *vegfab;vegfc* double mutants. Thus,

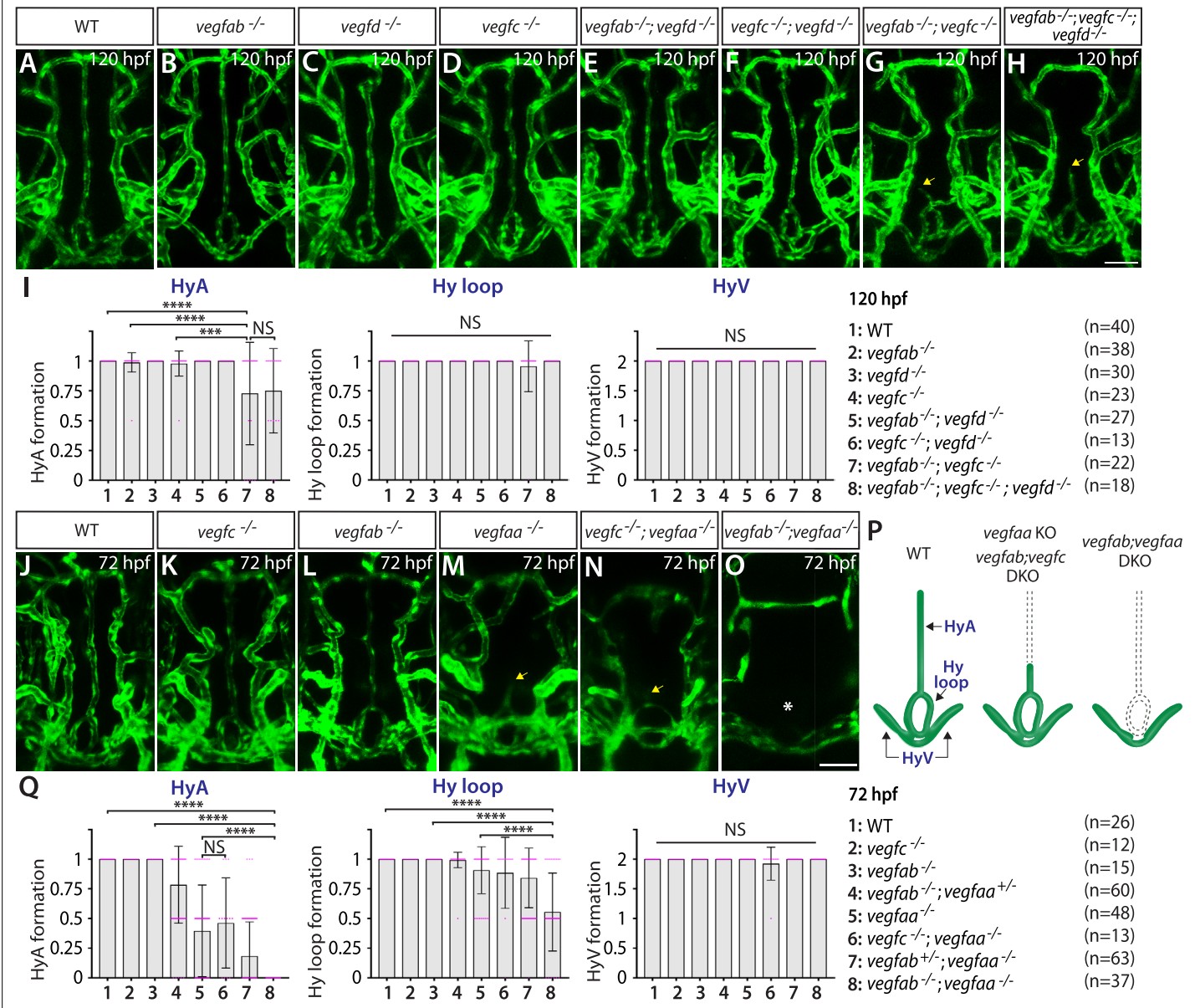

**Figure 7.** Heterogeneous endothelial requirements for Vegfs-dependent angiogenesis in the ventral brain around the neurohypophysis (NH)/organum vasculosum of the lamina terminalis (OVLT). (A–H) Dorsal views of 120 hours post fertilization (hpf) wild-type (WT) (A), *vegfab⁻/⁻* (B), *vegfd⁻/⁻* (C), *vegfc⁻/⁻* (D), *vegfab⁻/⁻;vegfd⁻/⁻* (E), *vegfc⁻/⁻;vegfd⁻/⁻* (F), *vegfab⁻/⁻;vegfc⁻/⁻* (G), and *vegfab⁻/⁻;vegfc⁻/⁻;vegfd⁻/⁻* (H) ventral brain vasculature visualized by *Tg(kdrl:EGFP)* expression. A significant fraction of *vegfab⁻/⁻;vegfc⁻/⁻* (G) and *vegfab⁻/⁻;vegfc⁻/⁻;vegfd⁻/⁻* (H) larvae exhibited a partial formation of the hypophyseal artery (HyA), resulting in HyA stalling toward the palatocerebral arteries (PLA) (arrows, G, H). (I) Quantification of HyA, Hy loop, and hypophyseal vein (HyV) formation at 120 hpf (the number of animals examined per genotype is listed in the panel). *vegfab⁻/⁻;vegfc⁻/⁻* and *vegfab⁻/⁻;vegfc⁻/⁻;vegfd⁻/⁻* larvae displayed a specific and partially penetrant defect in HyA formation. (J–O) Dorsal views of 72 hpf WT (J), *vegfc⁻/⁻* (K), *vegfab⁻/⁻* (L), *vegfaa⁻/⁻* (M), *vegfc⁻/⁻;vegfaa⁻/⁻* (N), and *vegfab⁻/⁻;vegfaa⁻/⁻* (O) ventral brain vasculature visualized by *Tg(kdrl:EGFP)* expression. Similar to *vegfab⁻/⁻;vegfc⁻/⁻* larvae, *vegfaa⁻/⁻* fish exhibited a specific and partially penetrant defect in HyA formation, leading to HyA stalling toward the PLA (arrow, M). The severity of this phenotype was exacerbated by the simultaneous deletion of *vegfab* (O), but not of *vegfc* (arrow, N), showing a genetic interaction between *vegfaa* and *vegfab*, but not between *vegfaa* and *vegfc*. Intriguingly, phenotypes in *vegfab⁻/⁻;vegfaa⁻/⁻* larvae were restricted to the fenestrated HyA and Hy loop (asterisk, O) with no significant defect in HyV formation. (P) Schematic representations of the severe vascular phenotypes observed in 72 hpf various *vegf* mutants. (Q) Quantification of HyA, Hy loop, and HyV formation at 72 hpf. In panels (I and Q), each data point shown in magenta represents individual animal's vessel formation score, and values represent means ± SD (*** and **** indicate p<0.001 and p<0.0001, respectively, by one-way analysis of variance [ANOVA] followed by Tukey's HSD test). Scale bars: 50 µm in (H) for (A–H) and in (O) for (J–O).

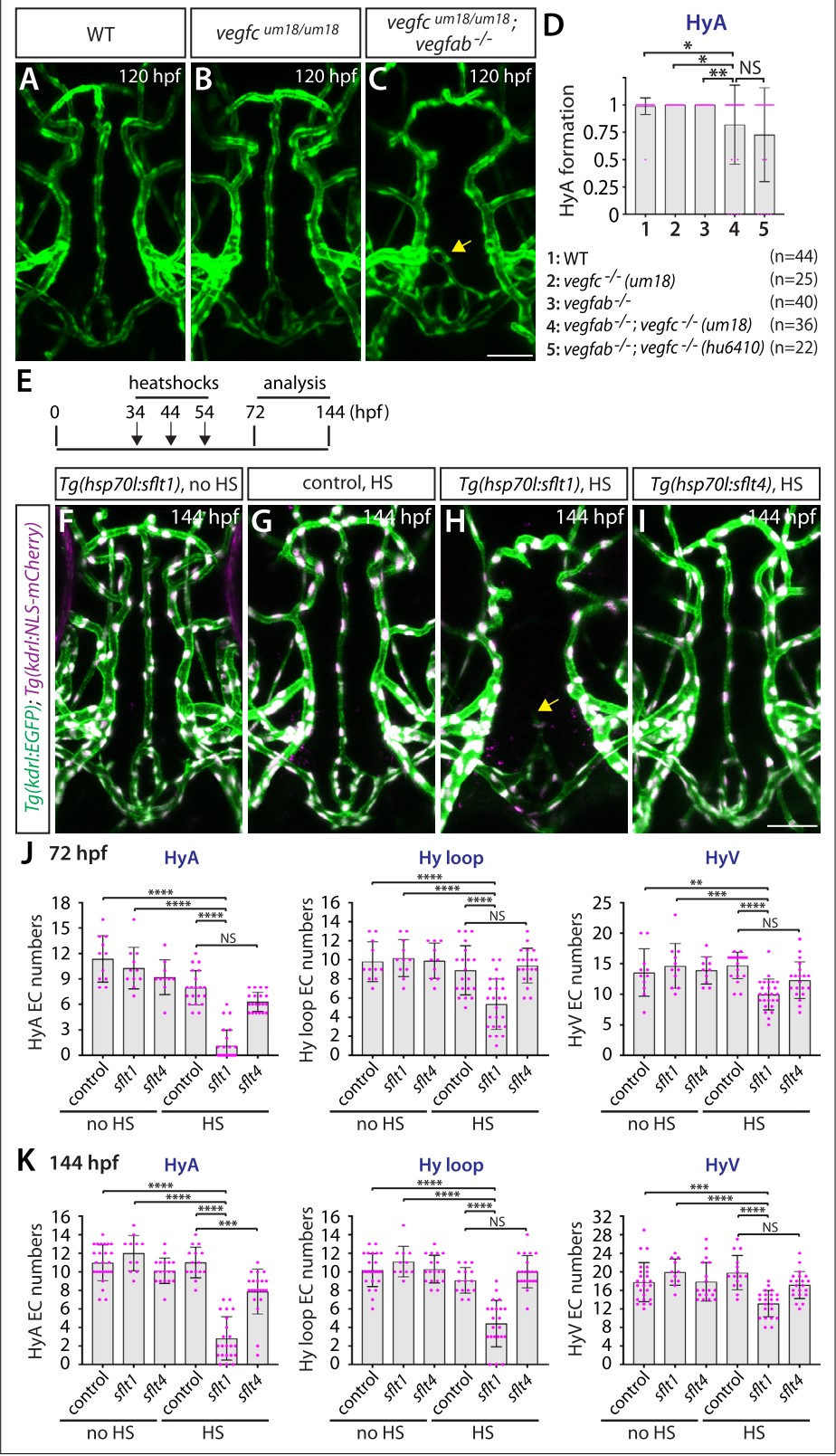

**Figure 8.** Temporal inhibition of Vegfa signaling by sFlt1 overexpression is sufficient to cause impaired formation of the hypophyseal artery (HyA) and Hy loop. (**A–C**) Dorsal views of 72 hours post fertilization (hpf) wild-type (WT) (**A**), *vegfc*[um18/um18] (**B**), and *vegfc*[um18/um18];*vegfab*[-/-] (**C**) ventral brain vasculature visualized by *Tg(kdrl*:EGFP) expression. *vegfc*[um18/um18];*vegfab*[-/-] larvae exhibited partial formation of the HyA, resulting in HyA stalling toward

*Figure 8 continued on next page*

*Figure 8 continued*

the palatocerebral arteries (PLA) (arrow, **C**). (**D**) Quantification of HyA formation at 120 hpf (the number of animals examined per genotype is listed in the panel). *vegfc*$^{um18/um18}$;*vegfab*$^{-/-}$ larvae exhibited a significantly increased defect in HyA formation at a comparable level to that observed in *vegfab*$^{-/-}$;*vegfc*$^{-/-}$ (*hu6410* allele) animals. The HyA quantitative results of *vegfab*$^{-/-}$;*vegfc*$^{-/-}$ (*hu6410* allele) larvae from **Figure 7I** were integrated in this graph for comparison purposes. (**E**) Time course of the heatshock (HS) experiments for panels (**F–I**). (**F–I**) Dorsal views of ventral brain vasculature in 144 hpf *Tg(hsp70l:sflt1)* (**F, H**), *Tg(hsp70l:sflt4)* (**I**), and their control sibling (**G**) larvae that carried both the *Tg(kdrl:EGFP)* and *Tg(kdrl:NLS-mCherry)* transgenes after treatment with (**G–I**) and without (**F**) multiple HS. HS-induced overexpression of sFlt1 caused severe defects in HyA formation, leading to HyA stalling toward the PLA (arrow, **H**). (**J, K**) Quantification of the number of vascular endothelial cells (vECs) that comprise the HyA, Hy loop, and hypophyseal vein (HyV) at 72 (**J**) and 144 (**K**) hpf. HS-induced overexpression of sFlt1 led to a drastic reduction in the number of vECs that comprise the HyA and milder reduction of vEC numbers in Hy loop. In contrast, sFlt4 overexpression displayed no effect on the number of vECs that comprise the Hy loop and HyV, but caused a significant reduction in HyA vEC numbers at 144 hpf (**K**). The number of animals examined per treatment at 72 hpf: without HS treatment, n=11 for control, n=11 for *sflt1*, and n=10 for *sflt4*; with HS treatment, n=20 for control, n=25 for *sflt1*, and n=20 for *sflt4*. The number of animals examined at 144 hpf: without HS treatment, n=25 for control, n=11 for *sflt1*, and n=18 for *sflt4*; with HS treatment, n=15 for control, n=22 for *sflt1*, and n=21 for *sflt4*. In panels (**D**), (**J**), and (**K**), each data point shown in magenta represents individual animal's quantification, and values represent means ± SD (*, **, ***, and **** indicate p<0.05, p<0.01, p<0.001, and p<0.0001, respectively, by one-way analysis of variance [ANOVA] followed by Tukey's HSD test). Scale bar: 50 μm in (**C**) for (**A–C**) and in (**I**) for (**F–I**).

temporal inhibition of Vegfa or Vegfc signaling during vascularization of the NH/OVLT is sufficient to cause HyA and Hy loop formation defects similarly to what was observed in the mutants.

Collectively, our results here present vessel type-selective angiogenesis. Vegfa paralogs and Vegfc selectively regulate the formation of strong *Tg(plvap:EGFP)*$^+$ Hy loop and the HyA with little contribution to faintly *Tg(plvap:EGFP)*$^+$ HyV development. These observations highlight local heterogeneity in endothelial requirements for Vegf-mediated angiogenesis.

## Normal choriocapillaris formation in zebrafish deficient for Wnt/β-catenin signaling, and conserved expression of Vegfa paralogs in retinal pigment epithelium

The choriocapillaris, or the choroidal vascular plexus (CVP), is a dense network of fenestrated capillaries that form in adjacent tissues outside of the retinal pigment epithelial cell (RPE) layer in the eyes (*Nickla and Wallman, 2010*; *Bill et al., 1983*; *Wybar, 1954*). This dense vascular network mediates efficient exchanges of nutrients and metabolic wastes in the outer retina (*Nickla and Wallman, 2010*; *Bill et al., 1983*; *Wybar, 1954*). Previous studies in mice indicate that *Vegfa* is expressed in RPE and is critical for fenestrated CVP formation (*Le et al., 2010*; *Saint-Geniez et al., 2006*; *Marneros et al., 2005*). In zebrafish, Vegfr2 receptor paralogs, Kdrl and Kdr, have been implicated to regulate CVP formation (*Ali et al., 2020*), however, it remains unclear which Vegf ligands are involved in this process. Given that the cellular and molecular mechanisms of CVP formation are better characterized than CP and CVO vascularization in mammals, we expected the CVP model to serve as a good system for addressing inter-species conservation of *vegf* expression and function in fenestrated CNS angiogenesis between mammals and zebrafish.

Similar to CP and CVO vasculature, we found strong *Tg(plvap:EGFP)* expression in the CVP at 6 dpf (*Figure 9A*) and noted endothelial fenestrations in this capillary network at the ultrastructure level at 10 dpf (*Figure 9B–D*). Confocal visualization of the CVP from the back of isolated eyes reveals no apparent defect in CVP formation in *gpr124*, *reck*, or *wnt7aa* mutants at 6 dpf (*Figure 9E–J*). These results suggest that Wnt/β-catenin signaling is not involved in fenestrated CVP formation similarly to what we found in fenestrated brain vascular beds (*Figures 1I-S and 2E-K*).

Next, we examined *vegf* expression using our BAC transgenic lines and found that at 54 hpf when active angiogenesis is occurring to form the CVP (*Ali et al., 2020*; *Hashiura et al., 2017*), both *vegfaa* and *vegfab* BAC reporter expression is co-labeled with an antibody for ZO-1, a tight junction protein marker for RPE (*Figure 9K–N'''*). Cryosections of these transgenic embryos at 50 hpf further confirmed strong and specific *vegfaa* and *vegfab* BAC reporter expression in the RPE layer within the outer retina, which lies adjacent to *Tg(kdrl:ras-mCherry)*$^+$ vECs comprising the CVP (*Figure 9S-V'''*).

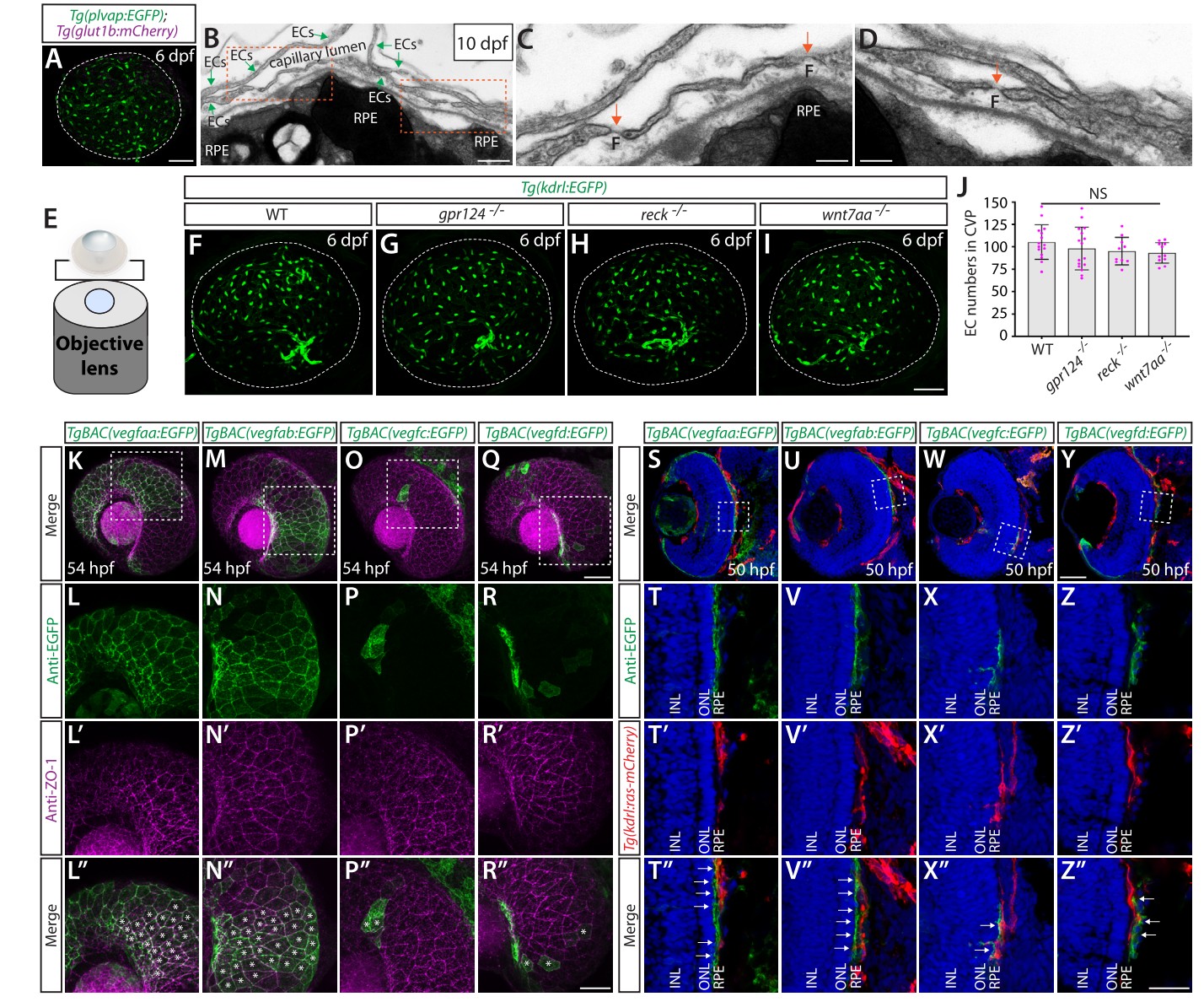

**Figure 9.** Normal choriocapillaris formation in zebrafish deficient for Wnt/β-catenin signaling, and conserved expression of Vegfa paralogs in retinal pigment epithelium (RPE). (**A**) Dissected eye from 10 days post fertilization (dpf) *Tg(plvap:EGFP);Tg(glut1b:mCherry)* zebrafish immunostained for GFP and DsRed shows strong *Tg(plvap:*EGFP) and absent *Tg(glut1b:*mCherry) expression in the choriocapillaris (choroidal vascular plexus, **CVP**). (**B–D**) Transmission electron microscopy images of 10 dpf wild-type (WT) outer retina focused on the CVP and RPE layer. Magnified images of the boxed areas in (**B**) show the presence of fenestrae (**F**) in the vascular endothelial cells (vECs) comprising the CVP (orange arrows, **C**, **D**). (**E**) Schematic diagram of 3D confocal CVP imaging from the back of dissected eyes. (**F–I**) WT (**F**), *gpr124⁻/⁻* (**G**), *reck⁻/⁻* (**H**), and *wnt7aa⁻/⁻* (**I**) CVP visualized by *Tg(kdrl:*EGFP) expression at 6 dpf. Confocal *z*-stack images of dissected eyes were taken after immunostaining for GFP. (**J**) Quantification of vECs that comprise the CVP at 6 dpf (n=15 for WT, n=16 for *gpr124⁻/⁻*, n=10 for *reck⁻/⁻*, and n=11 for *wnt7aa⁻/⁻* fish). No significant difference was observed across these genotypes. Each data point shown in magenta represents individual animal's quantification. Refer to *Figure 9—source data 1* for the precise cell counts of individual larvae. (**K–R″**) Lateral views of 54 hours post fertilization (hpf) *TgBAC(vegfaa:EGFP)* (**K**), *TgBAC(vegfab:EGFP)* (**M**), *TgBAC(vegfc:EGFP)* (**O**), and *TgBAC(vegfd:EGFP)* (**Q**) embryos immunostained for GFP and ZO-1, a tight junction marker for RPE. Magnified images of the boxed areas in (**K**), (**M**), (**O**), and (**Q**) are shown in (**L–L″**), (**N–N″**), (**P–P″**), and (**R–R″**), respectively. *TgBAC(vegfaa:*EGFP) and *TgBAC(vegfab:*EGFP) expression was broadly co-localized with ZO-1 immunoreactivity in RPE (asterisks in **L″**, **N″**). Sparse EGFP⁺ cells were observed in *TgBAC(vegfc:EGFP)* and *TgBAC(vegfd:EGFP)* eyes, some of which were co-localized with ZO-1 immunoreactivity (asterisks in **P″**, **R″**). (**S–Z″**) Cryosections of 50 hpf *TgBAC(vegfaa:EGFP)* (**S**), *TgBAC(vegfab:EGFP)* (**U**), *TgBAC(vegfc:EGFP)* (**W**), and *TgBAC(vegfd:EGFP)* (**Y**) embryos that carried the *Tg(kdrl:ras-mCherry)* transgene. Sections were immunostained for GFP and DsRed, and counterstained for DAPI. Magnified images of the boxed areas in (**S**), (**U**), (**W**), and (**Y**) are shown in (**T–T″**), (**V–V″**), (**X–X″**), and (**Z–Z″**), respectively. *TgBAC(vegfaa:*EGFP) and *TgBAC(vegfab:*EGFP) expression was broadly observed in the RPE layer directly adjacent to the CVP (white arrows in **T″**, **V″**). Sparse EGFP⁺ cells on *TgBAC(vegfc:EGFP)* and *TgBAC(vegfd:EGFP)* sections resided in

*Figure 9 continued on next page*

*Figure 9 continued*

close proximity to the CVP (white arrows in **X″**, **Z″**). **NL**: inner nuclear layer, **ONL**: outer nuclear layer. Scale bars: 500 nm in (**B**); 200 nm in (**C**), (**D**); 50 µm in (**A**), in (**I**) for (**F–I**), in (**Q**) for (**K**), (**M**), (**O**), in (**Y**) for (**S**), (**U**), (**W**); 30 µm in (**R″**) for (**L–L″**), (**N–N″**), (**P–P″**), (**R–R″**); 25 µm in (**Z″**) for (**T–T″**), (**V–V″**), (**X–X″**), (**Z–Z″**).

The online version of this article includes the following source data for figure 9:

**Source data 1.** Quantification of the number of endothelial cells that comprise the choroidal vascular plexus (CVP) in wild-type (WT), *gpr124*⁻/⁻, *reck*⁻/⁻, and *wnt7aa*⁻/⁻ at 6 days post fertilization (dpf).

Restricted *vegfaa* and *vegfab* expression to RPE is consistent with *Vegfa* expression patterns reported in developing murine retina (*Le et al., 2010*; *Saint-Geniez et al., 2006*). As compared to the broad expression of *vegfaa* and *vegfab* in RPE, *vegfc* and *vegfd* BAC reporter expression marked retinal cells sparsely, and a vast majority of RPE cells were devoid of reporter expression (*Figure 9O–R″ and W–Z″*).

## Functional redundancy of zebrafish Vegfa paralogs in CVP formation supports high conservation of fenestrated CNS angiogenic mechanisms between zebrafish and mammals

To determine which Vegf ligands are involved in CVP formation, we crossed mutants with *Tg(kdrl:EGFP);Tg(kdrl:NLS-mCherry)* double transgenic reporters that label endothelial cell bodies in EGFP and their nuclei in mCherry. These reporter lines enabled us to quantify the exact number of vECs that comprise the CVP. Since a previous time-lapse imaging study indicates that CVP vascular network is mostly formed by around 65 hpf (*Hashiura et al., 2017*), we chose to analyze mutants at 72 hpf or later. Based on prominent *vegfaa* and *vegfab* BAC reporter expression in RPE, we first examined mutants of these two genes by incrossing double heterozygous adults and analyzing their progeny at 72 hpf. Individual homozygous mutants displayed a considerable reduction in the number of vECs constituting the CVP, resulting in diminished vascular network elaboration, as compared to WT (*Figure 10A–C*). Homozygous *vegfaa* mutants showed underdeveloped eyes likely due to poor vascularization of this tissue. Importantly, we observed a strong genetic interaction between the two Vegfa paralogs, with increased deletion of either gene resulting in reduced numbers of vECs that comprise the CVP (*Figure 10A–E*). This finding indicates that both Vegfa paralogs are required for fenestrated CVP formation.

To inhibit Vegfa signaling in a temporally controlled manner, we employed the *Tg(hsp70l:sflt1)* line. Since active angiogenesis leading to CVP formation occurs between 36 and 65 hpf in zebrafish embryos (*Hashiura et al., 2017*), we subjected embryos to heatshocks at 34, 44, and 54 hpf, and analyzed them at 72 hpf (*Figure 10F*). Similar to what we observed in *vegfab* and *vegfaa* mutants, we found a drastic reduction in the number of vECs that comprise the CVP after sFlt1 overexpression (*Figure 10H–J and L*). This phenotype was not observed in *Tg(hsp70l:sflt4)* animals in which the Vegfc ligand trap, sFlt4, overexpression was induced under the same heatshock regimen (*Figure 10K and L*). To eliminate the possibility of potential developmental delays in heatshock-treated *Tg(hsp70l:sflt1)* larvae, we also analyzed them at 144 hpf, following additional heatshocks every 24 hr after 54 hpf (*Figure 10G*). The results were similar even at this later stage (*Figure 10M*), demonstrating that inhibition of Vegfa-induced signaling during CVP formation is sufficient to abrogate this process.

The results of heatshock-treated *Tg(hsp70l:sflt4)* larvae indicate that Vegfc and/or Vegfd themselves do not contribute to CVP formation. Nevertheless, we analyzed these mutants individually and in combinations to determine their definite contributions to CVP formation. We observed that *vegfc* and *vegfd* single mutants or *vegfc;vegfd* double mutants showed either no significant difference or a slight reduction in the number of vECs that comprise the CVP at 6 dpf (*Figure 10R*). Moreover, we found no genetic interactions between *vegfab* and *vegfd* or *vegfc* in CVP formation (*Figure 10N–R*). This finding, combined with the lack of significant changes in vEC numbers of the CVP following sFlt4 overexpression, suggests that Vegfc or Vegfd contributions to CVP formation are minor, if any.

Together, our observations here reveal that the combined deletion of *vegfab*, *vegfc*, and *vegfd* does not increase the severity of defects in fenestrated CVP formation compared to *vegfab* single mutants. Instead, *vegfab* and *vegfaa* display a strong genetic interaction in CVP formation, which is further supported by the results from sFlt1-mediated temporal Vegfa inhibition experiments (*Figure 10S*).

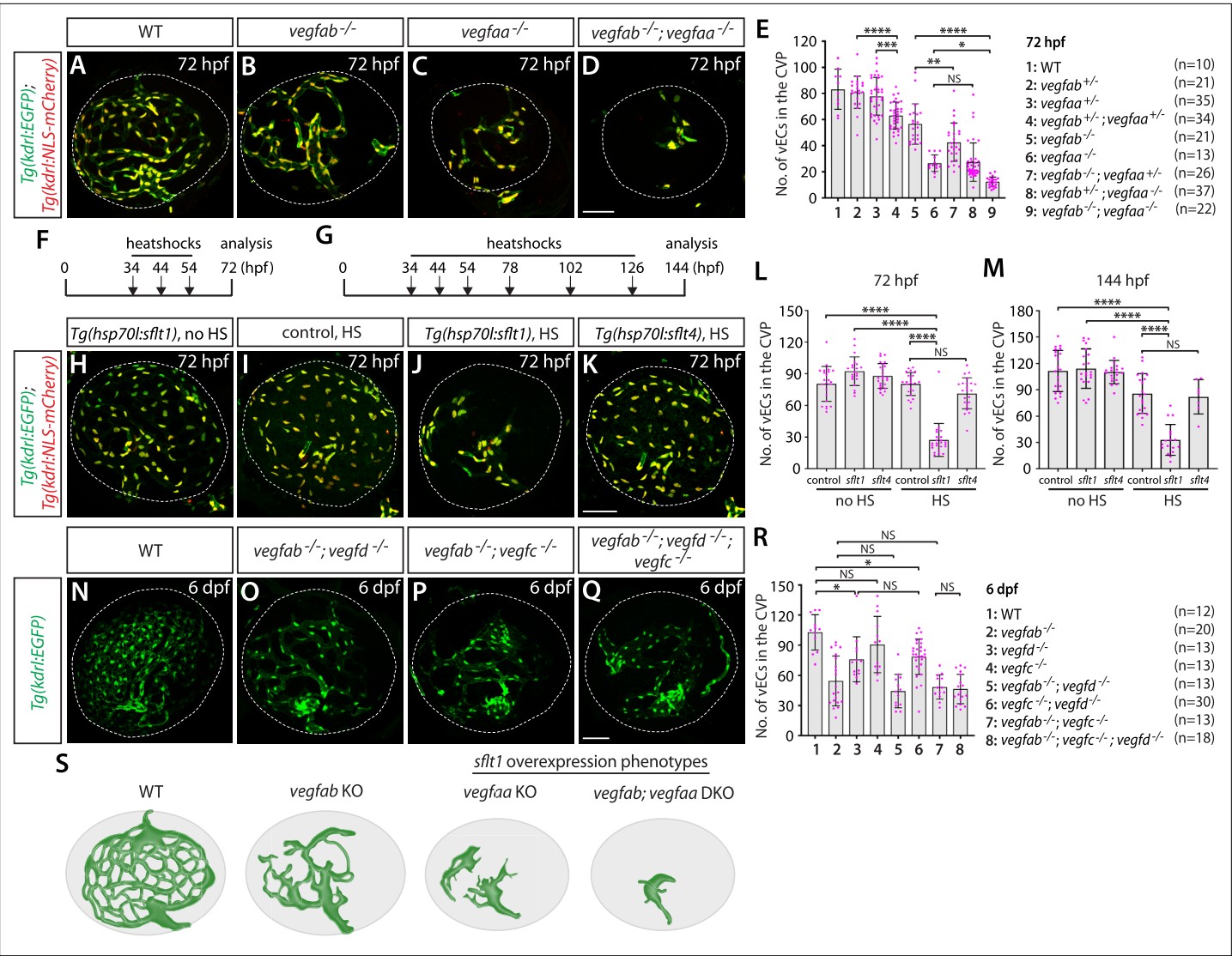

**Figure 10.** Zebrafish Vegfa paralogs are redundantly required for fenestrated choroidal vascular plexus (CVP) formation. (**A–D**) Wild-type (WT) (**A**), *vegfab⁻/⁻* (**B**), *vegfaa⁻/⁻* (**C**), and *vegfab⁻/⁻;vegfaa⁻/⁻* (**D**) CVP visualized by *Tg(kdrl:*EGFP) and *Tg(kdrl:*NLS-mCherry) expression at 72 hours post fertilization (hpf). (**E**) Quantification of the number of vascular endothelial cells (vECs) that comprise the CVP at 72 hpf (the number of animals examined per genotype is listed in the panel). Zebrafish *vegfa* paralogs genetically interacted in fenestrated CVP formation. (**F, G**) Time course of the heatshock (HS) experiments for panels (**H–M**). (**H–K**) The CVP of 72 hpf *Tg(hsp70l:sflt1)* (**H, J**), *Tg(hsp70l:sflt4)* (**K**), and their control sibling (**I**) larvae that carried both the *Tg(kdrl:EGFP)* and *Tg(kdrl:NLS-mCherry)* transgenes after treatment with (**I–K**) and without (**H**) multiple HS. HS-induced overexpression of sFlt1 led to pronounced reductions in the number of vECs constituting the CVP (**J**). (**L, M**) Quantification of the number of vECs that comprise the CVP at 72 (**L**) and 144 (**M**) hpf with and without HS treatments. The number of animals examined per treatment at 72 hpf: without HS treatment, n=23 for control, n=21 for *sflt1*, and n=25 for *sflt4*; with HS treatment, n=23 for control, n=24 for *sflt1*, and n=23 for *sflt4*. The number of animals examined at 144 hpf: without HS treatment, n=22 for control, n=21 for *sflt1*, and n=20 for *sflt4*; with HS treatment, n=20 for control, n=28 for *sflt1*, and n=24 for *sflt4*. (**N–Q**) WT (**N**), *vegfab⁻/⁻;vegfd⁻/⁻* (**O**), *vegfab⁻/⁻;vegfc⁻/⁻* (**P**), and *vegfab⁻/⁻;vegfd⁻/⁻;vegfc⁻/⁻* (**Q**) CVP visualized by *Tg(kdrl:EGFP)* expression at 6 days post fertilization (dpf). (**R**) Quantification of the number of vECs that comprise the CVP at 6 dpf (the number of animals examined per genotype is listed in the panel). Refer to *Figure 10—source data 1* for the precise cell counts of individual larvae. (**S**) Schematic representations of the CVP phenotypes observed in *vegfa* mutants and after sFlt1 overexpression at 72 hpf. Temporal inhibition of Vegfa signaling by sFlt1 overexpression recapitulated the severely impaired CVP phenotypes observed in genetic mutants. In panels (**E**), (**L**), (**M**), and (**R**), each data point shown in magenta represents individual animal's quantification, and values represent means ± SD (*, **, ***, and **** indicate p<0.05, p<0.01, p<0.001, and p<0.0001, respectively, by one-way analysis of variance [ANOVA] followed by Tukey's HSD test). Scale bars: 50 µm in (**D**) for (**A–D**), in (**K**) for (**H–K**), in (**Q**) for (**N–Q**).

The online version of this article includes the following source data for figure 10:

**Source data 1.** Quantification of the number of endothelial cells that comprised the choroidal vascular plexus (CVP) in wild-type (WT) and various *vegf* mutants at 6 days post fertilization (dpf).

Combined with *vegfab* and *vegfaa* restricted expression in RPE, these data suggest well-conserved cellular and molecular mechanisms underlying CVP formation between zebrafish and mammals. Moreover, these results provide further evidence for inter-tissue heterogeneity of angiogenic regulation across fenestrated CNS vascular beds since the combined deletion of *vegfaa* and *vegfab* caused no fenestrated vessel formation defect at the dCP/PG interface (*Figure 4N*).

## Discussion

Generation of both fenestrated and BBB-forming capillaries allows diverse neural activities and is vital to brain homeostasis. However, in contrast to recent advances in our understanding of BBB angiogenic mechanisms, fenestrated capillary development in the CPs and CVOs is still poorly understood. Here, we report substantial functional redundancy among Vegf ligands required for vascularization of multiple fenestrated CNS vascular beds. We show that the individual loss of four different Vegf ligands (Vegfaa, Vegfab, Vegfc, and Vegfd) in zebrafish causes either undetectable or partially penetrant defects in fenestrated vessel formation in the dCP/PG, NH/OVLT, and retinal choroid. However, the simultaneous loss of specific Vegf ligand combinations results in dramatically enhanced angiogenic defects in a vessel- and organ-specific manner. Comparative analysis across these CNS regions enabled us to identify common and CNS region-specific Vegf requirements for vascularization (*Figure 11*). In particular, our findings suggest the great importance of Vegfc/d and Vegfa interplay in angiogenesis leading to fenestrated brain vascular beds. Expression analysis using BAC transgenic

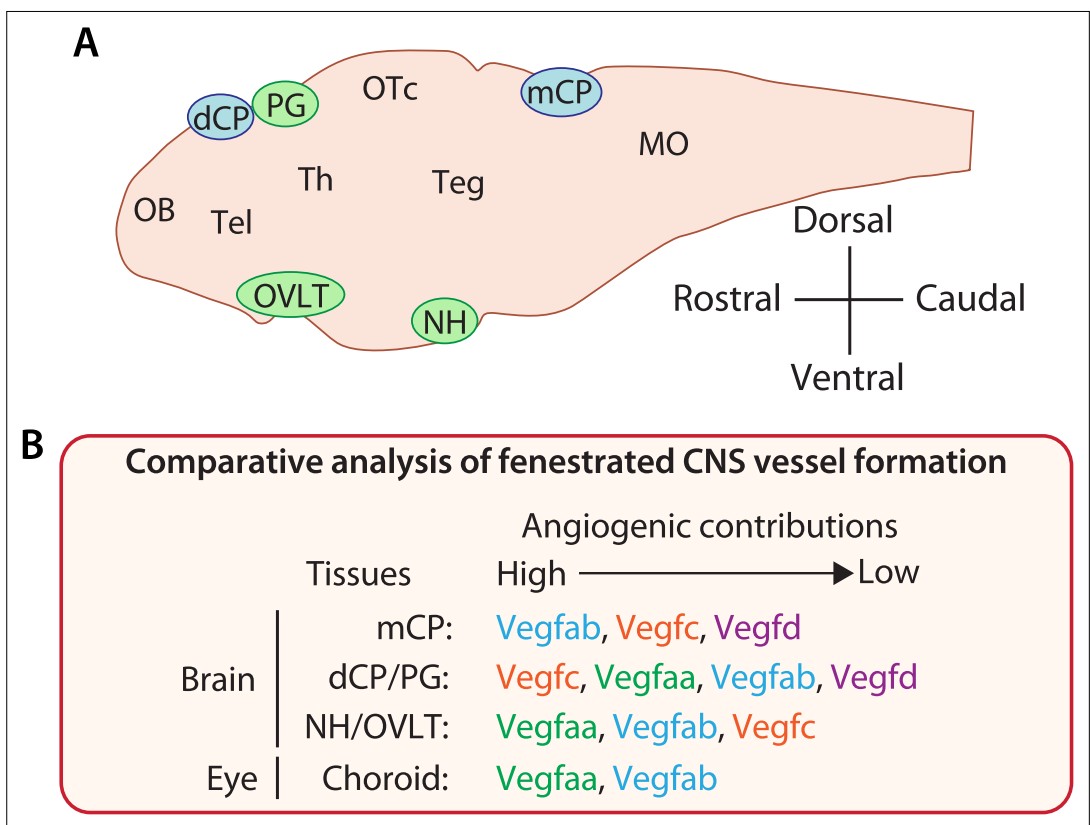

**Figure 11.** Comparative analysis of vascularization across fenestrated central nervous system (CNS) vascular beds reveals crucial angiogenic interplay of Vegfc/d and Vegfa in choroid plexus (CP) and circumventricular organ (CVO) vascularization. (**A**) Schematic diagram of a mid-sagittal section of the larval zebrafish brain indicates the location of the CPs and CVOs examined in our current and previous studies. MO: medulla oblongata, OB: olfactory bulb, OTc: optic tectum, Teg: tegmentum, Tel: telencephalon, Th: thalamus. (**B**) Summary of findings from our comparative analysis of fenestrated vessel formation. Angiogenic contributions were speculated based on the severity and penetrance of observed phenotypes and also on the levels of genetic interactions detected to cause phenotypes. This comparative result reveals CNS region-specific requirements for Vegfs-dependent angiogenesis and identifies an unexpected role for Vegfc/d in fenestrated brain vessel formation.

reporters supported the genetic results and revealed partially overlapping expression domains of *vegf* genes during vessel morphogenesis in each of these CNS regions. We observed notable *vegf* expression in extending vECs and/or non-neuronal cell types that are uniquely present in the CPs, CVOs, and retina, including epithelial cells in the CPs, pituicytes in the NH, and RPE in the outer retina. These expression patterns indicate crucial cellular sources of Vegfs that enable their locally restricted interplay in CP and CVO vascularization. Genetic analysis of paracrine activity-deficient *vegfc*[um18] mutants implies brain region-specific requirements of endothelial cell-autonomous and cell non-autonomous Vegfc angiogenic function in fenestrated vessel formation. Lastly, RPE-dominant expression of zebrafish *vegfa* paralogs and their functional redundancy in CVP formation supports high conservation of the cellular and molecular mechanisms underlying fenestrated capillary formation between the zebrafish and mammalian CNS.

## Capillary type-selective brain angiogenesis directed by Vegfs and Wnt7/β-catenin signaling

How phenotypically heterogeneous networks of brain vasculature arise remains largely unclear. We recently reported that a redundant angiogenic activity of Vegfab, Vegfc, and Vegfd is restricted to fenestrated mCP vascular development with little impact on the formation of the neighboring BBB vasculature (*Parab et al., 2021*). Conversely, we showed here that the genetic loss of Wnt7/β-catenin signaling (*gpr124*, *reck*, or *wnt7aa* KO) causes a severe angiogenesis defect in the brain parenchyma without any apparent deficit in fenestrated vasculature across the CNS. This observation is in line with the recent studies in mice, which reported low levels of β-catenin activation in fenestrated vascular beds of the CPs and CVOs compared to those that form the BBB (*Wang et al., 2019*; *Benz et al., 2019*). In *vegfab*[-/-];*vegfaa*[-/-] larvae, we further observed a selective deficit in BBB markers[+] MsV formation with little effect on the neighboring fenestrated vasculature at the dCP/PG interface, presenting another striking example of capillary type-specific angiogenesis. Notably, *vegfab*[-/-];*vegfaa*[-/-] larvae displayed pronounced defects in fenestrated Hy loop and HyA formation in the NH/OVLT regions, demonstrating heterogeneous requirements of the same Vegf ligand combination for BBB and fenestrated angiogenesis across organs.

Our mutant analysis provides compelling genetic evidence that CP and CVO vascularization with fenestrated capillaries is independent of Wnt7/Gpr124/Reck signaling. Conversely, BBB angiogenesis, including hindbrain CtA formation, requires Vegfa signaling as previously documented (*Fetsko et al., 2023*; *Rossi et al., 2016*; *Lange et al., 2022*). Adding to these prior observations, our results suggest the central role of Vegfa signaling in BBB markers[+] MsV formation. Thus, BBB angiogenesis is not completely independent of Vegf signaling. However, in light of our findings, we expect Vegfa and Vegfc/d interplay to be highly specific to fenestrated vessel formation within the restricted brain regions we examined around the CPs and CVOs. Further investigations will be needed to determine if this interplay is also required for induction of endothelial fenestration and high permeability similarly to the requirement of Wnt/β-catenin signaling for BBB angiogenesis and barrier properties (*Langen et al., 2019*; *Daneman and Prat, 2015*).

## Local and inter-tissue heterogeneity of Vegfs-dependent angiogenesis during the vascularization of fenestrated CNS vascular beds

Current evidence suggests that the development of complex brain vascular networks requires extensive functional redundancy among signaling pathways. For example, brain vasculature that forms the BBB is not uniformly compromised in mice deficient for Wnt/β-catenin signaling (*Daneman et al., 2009*; *Stenman et al., 2008*; *Wang et al., 2018*). This is because Wnt ligands (Wnt7a, Wnt7b, Norrin) and their receptors display significant redundancy in BBB formation/maintenance in specific brain regions (*Cho et al., 2017*; *Daneman et al., 2009*; *Stenman et al., 2008*; *Wang et al., 2018*; *Zhou and Nathans, 2014*; *Zhou et al., 2014*). Wnt7a and Wnt7b are redundantly required for the vascularization of the ventral neural tube in mice (*Daneman et al., 2009*; *Stenman et al., 2008*). Wnt7a/b and Norrin genetically interact in regulating BBB maintenance in the cerebellum (*Wang et al., 2018*). These prior studies indicate brain region-specific molecular requirements for vascularization and vEC barrier properties.

In this study, we began by asking whether the combination of Vegf ligands (Vegfab, Vegfc, and Vegfd) required for mCP vascularization is the master regulator of fenestrated capillary formation

across the brain. Our comparative analysis of the CPs and CVOs revealed highly heterogeneous molecular requirements for angiogenesis within and across these organs. This local and inter-tissue heterogeneity of endothelial requirements for Vegfs-dependent angiogenesis is supported by brain area-specific, unique gene expression of these angiogenic factors. In all the brain regions we examined, we observed partially overlapping *vegf* expression patterns in either developing vECs or non-neuronal cell types that specifically reside in/around the CPs or CVOs.

How are fenestrated angiogenic programs encoded in a brain region-specific manner? In the zebrafish NH, specialized astroglia, pituicytes, were implicated as a potential source of Vegfab and Vegfaa, which maintain high vascular permeability in this region (*Anbalagan et al., 2018*), consistent with our observations of these two genes' expression in pituicytes at early larval stages. In the CPs of adult mice, epithelial cells express high levels of Vegfa (*Maharaj et al., 2008*) that maintain the fenestrated states of vasculature in this region (*Maharaj et al., 2008*; *Kamba et al., 2006*). Our present study suggests that epithelial cells express *vegfab* and *vegfc* in the developing dCP, which together direct the vascularization of the CP/PG interface. Given these observations, we speculate that non-neuronal cell types present in the CPs and CVOs build unique endothelial features to meet specific needs in these brain regions. Additionally, it is a possibility that brain region-specific angiogenic modulators are present to shape spatiotemporal activities of redundant angiogenic cues.

## Identification of Vegfc/d as crucial regulators of fenestrated brain vessel formation

An unexpected finding made through our comparative analysis is that fenestrated vessel formation requires Vegfc/d across multiple brain regions. In contrast to the well-characterized function of VEGFA in angiogenesis, endothelial fenestration, and vascular permeability both in vitro and in vivo (*Apte et al., 2019*; *Maharaj and D'Amore, 2007*; *Matsuoka and Stainier, 2018*), Vegfc/d function in this context is largely unclear in vivo. Furthermore, the role of Vegfc/d in brain vascularization has not been well established except the previous studies reporting that *vegfc* loss-of-function leads to a transient, partially penetrant defect in the formation of the primordial hindbrain channel in early embryonic stages of zebrafish (*Villefranc et al., 2013*; *Covassin et al., 2006*) and our recent work on revealing Vegfc/d's contributions to mCP vascularization (*Parab et al., 2021*). Nevertheless, the expression of VEGFC and its major receptor VEGFR3 in multiple neuroendocrine organs and their fenestrated capillaries was documented in human tissues (*Partanen et al., 2000*). Consistent with this report, our *vegfc* BAC transgenic zebrafish reporter marks developing fenestrated capillaries in addition to cells that reside within several neuroendocrine organs, including the pineal and pituitary gland. These observations imply a conserved function of Vegfc in regulating fenestrated capillary formation/integrity across vertebrates.

Our findings raise many questions that remain unanswered. We noted that *vegfc* displays a strong genetic interaction with *vegfa* paralog(s) to drive fenestrated vessel formation across the CPs and CVOs. Does Vegfc play a crucial role in endothelial fenestration development and/or maintenance? Does the redundancy between Vegfc and Vegfa paralog(s) occur at the level of their receptor(s) and downstream signaling pathways, or through a genetic compensatory mechanism? Is proteolytic activation of Vegfc involved in fenestrated brain vessel development? If so, what are these proteolytic regulators and where are they expressed? Is Vegfc function in this context conserved in mammals? Addressing these questions will further advance our understanding of Vegfc function in brain vascularization and barrier properties. Although not addressed in this study, there are additional Vegf family members, including Vegfba and Vegfbb (*Jensen et al., 2015*), which may account for partially penetrant brain vascular phenotypes observed in zebrafish mutants lacking Vegfa and Vegfc/d. Previous work conducting morpholino-mediated *vegfba* knockdown in zebrafish implied the potential role of Vegfba in regulating brain angiogenesis (*Jensen et al., 2015*). It will be interesting to further determine Vegfb contributions to developmental angiogenesis leading to fenestrated brain vascular beds.

Another important question includes which receptors mediate angiogenic activities evoked by multiple Vegf ligands during the vascularization of the CPs and CVOs. In zebrafish, the Vegf receptor family consists of four members: Flt1, Kdr, Kdrl, and Flt4 (*Covassin et al., 2006*; *Bussmann et al., 2008*; *Bahary et al., 2007*). Biochemical and genetic studies suggest that each Vegf ligand can bind to multiple Vegf receptors (*Vogrin et al., 2019*) and that these receptors are expressed in a partially overlapping manner during development (*Covassin et al., 2006*; *Bahary et al., 2007*; *Vogrin et al.,*

2019) and interact genetically in regulating developmental angiogenesis and lymphangiogenesis (*Covassin et al., 2006*; *Bahary et al., 2007*; *Vogrin et al., 2019*; *Quick et al., 2021*). Given the high degree of the functional redundancy among Vegf ligands required for CP and CVO vascularization, it is likely that multiple receptors are also involved redundantly in mediating this process. Future investigations will be needed to identify specific receptors and signaling pathways downstream of the Vegf ligands responsible for CP and CVO vascularization.

In summary, defining unique sets of molecular cues required for the formation and maintenance of specialized vascular beds within the brain is fundamental to designing vascular bed-specific cerebrovascular therapies. Given a wide range of organs that form fenestrated vasculature outside of the CNS, including in endocrine glands (pancreas and thyroid), kidney, and intestine, our findings may have broad impacts on our understanding of fenestrated endothelial development and specializations beyond the CNS. Finally, to extend our work in zebrafish, future investigation of the Vegfc/d signaling axis as a potential modulator of fenestrated vessel formation in mammals is warranted.

# Materials and methods

**Key resources table**

| Reagent type (species) or resource | Designation | Source or reference | Identifiers | Additional information |
|---|---|---|---|---|
| Genetic reagent (*Danio rerio*) | Tg(kdrl:EGFP)$^{s843}$ | *Jin et al., 2005* | ZFIN: s843 | |
| Genetic reagent (*Danio rerio*) | Tg(kdrl:Has.HRAS-mcherry)$^{s896}$ | *Chi et al., 2008* | ZFIN: s896 | |
| Genetic reagent (*Danio rerio*) | Tg(kdrl:NLS-mCherry)$^{is4}$ | *Wang et al., 2010* | ZFIN: is4 | |
| Genetic reagent (*Danio rerio*) | Tg(UAS:EGFP)$^{nkuasgfp1a}$ | *Asakawa et al., 2008* | ZFIN: nkuasgfp1a | |
| Genetic reagent (*Danio rerio*) | Tg(UAS:EGFP-CAAX)$^{m1230}$ | *Fernandes et al., 2012* | ZFIN: m1230 | |
| Genetic reagent (*Danio rerio*) | Tg(hsp70l:sflt1, cryaa-cerulean)$^{bns80}$ | *Matsuoka et al., 2016* | ZFIN: bns80 | |
| Genetic reagent (*Danio rerio*) | Tg(hsp70l:sflt4, cryaa-cerulean)$^{bns82}$ | *Matsuoka et al., 2016* | ZFIN: bns82 | |
| Genetic reagent (*Danio rerio*) | TgBAC(vegfab:gal4ff)$^{bns273}$ | *Mullapudi et al., 2019* | ZFIN: bns273 | |
| Genetic reagent (*Danio rerio*) | TgBAC(vegfaa:gal4ff)$^{lri96}$ | *Parab et al., 2021* | ZFIN: lri96 | |
| Genetic reagent (*Danio rerio*) | TgBAC(vegfc:gal4ff)$^{bns270}$ | *Parab et al., 2021* | ZFIN: bns270 | |
| Genetic reagent (*Danio rerio*) | TgBAC(vegfd:gal4ff)$^{lri95}$ | *Parab et al., 2021* | ZFIN: lri95 | |
| Genetic reagent (*Danio rerio*) | Tg(glut1b:mCherry)$^{sj1}$ | *Umans et al., 2017* | ZFIN: sj1 | |
| Genetic reagent (*Danio rerio*) | Tg(plvapb:EGFP)$^{sj3}$ | *Umans et al., 2017* | ZFIN: sj3 | |
| Genetic reagent (*Danio rerio*) | Et(cp:EGFP)$^{sj2}$ | *Henson et al., 2014* | ZFIN: sj2 | |
| Genetic reagent (*Danio rerio*) | Tg(fli1:GAL4-Hsa.TCF7L2-2A-mCherry)$^{ncv15}$ | *Kashiwada et al., 2015* | ZFIN: ncv15 | |
| Genetic reagent (*Danio rerio*) | Tg(fli1:MYR-mCherry)$^{ncv1}$ | *Kwon et al., 2013* | ZFIN: ncv1 | |
| Genetic reagent (*Danio rerio*) | vegfaa$^{bns1}$ | *Rossi et al., 2016* | ZFIN: bns1 | |

*Continued on next page*

| Reagent type (species) or resource | Designation | Source or reference | Identifiers | Additional information |
|---|---|---|---|---|
| Genetic reagent (*Danio rerio*) | *vegfab*<sup>bns92</sup> | *Rossi et al., 2016* | ZFIN: bns92 | |
| Genetic reagent (*Danio rerio*) | *vegfc*<sup>hu6410</sup> | *Helker et al., 2013* | ZFIN: hu6410 | |
| Genetic reagent (*Danio rerio*) | *vegfc*<sup>um18</sup> | *Villefranc et al., 2013* | ZFIN: um18 | |
| Genetic reagent (*Danio rerio*) | *vegfd*<sup>bns257</sup> | *Gancz et al., 2019* | ZFIN: bns257 | |
| Genetic reagent (*Danio rerio*) | *gpr124*<sup>s984</sup> | *Vanhollebeke et al., 2015* | ZFIN: s984 | |
| Genetic reagent (*Danio rerio*) | *wnt7aa*<sup>ulb2</sup> | *Martin et al., 2022* | ZFIN: ulb2 | |
| Genetic reagent (*Danio rerio*) | *reck*<sup>ulb3</sup> | This manuscript | ZFIN: ulb3 | |
| Antibody | Anti-GFP (chicken polyclonal) | Aves Labs | Cat#: GFP-1010 | IF (1:1000) |
| Antibody | Anti-GFP (rabbit polyclonal) | Thermo Fisher Scientific | Cat#: A-11122 | IF (1:200) |
| Antibody | Anti-GFP (chicken polyclonal) | Thermo Fisher Scientific | Cat#: A10262 | IF (1:200) |
| Antibody | Anti-DsRed (rabbit polyclonal) | Clontech Labs | Cat#: 632496 | IF (1:300) |
| Antibody | Anti-Claudin-5 (mouse monoclonal) | Thermo Fisher Scientific | Cat#: 35–2500 | IF (1:500) |
| Antibody | Anti-ZO-1 (mouse monoclonal) | Thermo Fisher Scientific | Cat#: 33–9100 | IF (1:500) |
| Antibody | Anti-P-glycoprotein (mouse monoclonal) | Thermo Fisher Scientific | Cat#: MA1-26528 | IF (1:100) |
| Antibody | Anti-Glut1 (rabbit polyclonal) | Novus Biologicals | Cat#: NB300-666 | IF (1:200) |
| Sequence-based reagent | *reck* sgRNA | This paper | Single-guide RNA | CCTGACAGTACTCACGAC |
| Commercial assay or kit | MEGAshortscript T7 Transcription | Thermo Fisher Scientific | Cat#: AM1354 | |
| Commercial assay or kit | mMessage mMachine T3 Transcription Kit | Thermo Fisher Scientific | Cat#: AM1348 | |
| Commercial assay or kit | DIG RNA Labeling Kit | MilliporeSigma | Cat#: 11277073910 | |
| Chemical compound, drug | Dextran, Tetramethylrhodamine, 3000 MW | Thermo Fisher Scientific | Cat#: D3308 | |
| Chemical compound, drug | Dextran, Alexa Fluor 647; 10,000 MW | Thermo Fisher Scientific | Cat#: D22914 | |
| Software, algorithm | LAS X Version 3.7.0.20979 | Leica Microsystems | | |
| Software, algorithm | ZEN 2.3 Blue Edition | Carl Zeiss Microscopy | | |
| Software, algorithm | GraphPad Prism 8.1.1 | GraphPad Software | | |
| Software, algorithm | Adobe Photoshop CS6 | Adobe | | |
| Software, algorithm | Adobe Illustrator CS6 | Adobe | | |

## Zebrafish husbandry and strains

All zebrafish husbandry was performed under standard conditions in accordance with institutional and national ethical and animal welfare guidelines. All zebrafish work was approved by the Cleveland Clinic's Institutional Animal Care and Use Committee under the protocol number 00002684. The following lines were used in this study: *Tg(kdrl:EGFP)*<sup>s843</sup> (*Jin et al., 2005*); *Tg(kdrl:Has.HRAS-mcherry)*<sup>s896</sup> (*Chi et al., 2008*), abbreviated *Tg(kdrl:ras-mCherry)*; *Tg(kdrl:NLS-mCherry)*<sup>is4</sup> (*Wang et al., 2010*); *Tg(UAS:EGFP)*<sup>nkuasgfp1a</sup> (*Asakawa et al., 2008*); *Tg(UAS:EGFP-CAAX)*<sup>m1230</sup> (*Fernandes et al.,*

*2012*); *Tg(hsp70l:sflt1, cryaa-cerulean)*[bns80] (*Matsuoka et al., 2016*), abbreviated *Tg(hsp70l:sflt1)*; *Tg(hsp70l:sflt4, cryaa-cerulean)*[bns82] (*Matsuoka et al., 2016*), abbreviated *Tg(hsp70l:sflt4)*; *TgBAC(vegfab:gal4ff)*[bns273] (*Mullapudi et al., 2019*); *TgBAC(vegfaa:gal4ff)*[lri96] (*Parab et al., 2021*); *TgBAC(vegfc:gal4ff)*[bns270] (*Parab et al., 2021*); *TgBAC(vegfd:gal4ff)*[lri95] (*Parab et al., 2021*); *Tg(glut1b:mCherry)*[sj1] (*Umans et al., 2017*); *Tg(plvapb:EGFP)*[sj3] (*Umans et al., 2017*), abbreviated *Tg(plvap:EGFP)*; *Et(cp:EGFP)*[sj2] (*Henson et al., 2014*); *Tg(fli1:GAL4-Hsa.TCF7L2-2A-mCherry)*[ncv15] (*Kashiwada et al., 2015*); *Tg(fli1:MYR-mCherry)*[ncv1] (*Kwon et al., 2013*); *vegfaa*[bns1] (*Rossi et al., 2016*); *vegfab*[bns92] (*Rossi et al., 2016*); *vegfc*[hu6410] (*Helker et al., 2013*); *vegfc*[um18] (*Villefranc et al., 2013*); *vegfd*[bns257] (*Gancz et al., 2019*); *gpr124*[s984] (*Vanhollebeke et al., 2015*); and *wnt7aa*[ulb2] (*Martin et al., 2022*). Adult fish were maintained on a standard 14 hr light/10 hr dark daily cycle. Fish embryos/larvae were raised at 28.5°C. To prevent skin pigmentation, 0.003% phenylthiourea (PTU) was used beginning at 10–12 hpf for imaging. Fish larvae analyzed at 10 dpf were transferred to a tank containing approximately 250 mL system water supplemented with 0.003% PTU (up to 25 larvae/tank) and fed with Larval AP100 (<50 µm dry diet, Zeigler) starting at 5 dpf.

## Genotyping of mutants

Genotyping of *vegfaa*[bns1] and *vegfab*[bns92] mutant fish was performed by high-resolution melt analysis (HRMA) of PCR products as described previously (*Parab et al., 2021*). Genotyping of *vegfc*[hu6410], *vegfc*[um18], and *vegfd*[bns257] mutant fish was performed by standard PCR as described previously (*Parab et al., 2021*; *Gancz et al., 2019*). Genotyping of *gpr124*[s984] and *wnt7aa*[ulb2] mutant fish was performed by HRMA of PCR products using the following primers:

> *gpr124 s984* forward: 5' – AGGGTCCACTGGAACTGCACACATTG – 3'
> *gpr124 s984* reverse 5' – TGCAATGGAAAGGCAGCCTGTCTCC – 3'
> *wnt7aa ulb2* forward: 5' – GCGCAAATGGGAATCAATGAGTG – 3'
> *wnt7aa ulb2* reverse: 5' – TCCAAAGACAGTTCTTTCCCCGAG – 3'

## Generation and genotyping of *reck* mutants

The zebrafish *reck*[ulb3] allele was engineered by targeted genome editing using the CRISPR/Cas9 system, as described previously (*Martin et al., 2022*; *America et al., 2022*). sgRNA target site was designed using the CRISPR Design website (http://crispr.mit.edu). The oligos (5' – taggCCTGACAGTACTCACGAC – 3' and 5' – aaacGTCGTGAGTACTGTCAGG – 3') were annealed and ligated into the pT7-gRNA vector (#46759, Addgene) (*Jao et al., 2013*) after digesting the plasmid with BsmBI (NEB). The sgRNA was transcribed from the BamHI linearized pT7-gRNA vector using the MEGAshortscript T7 kit (#AM1354, Thermo Fisher Scientific). The synthetic Cas9 mRNA was transcribed from the XbaI linearized pT3TS-nls-zCas9-nls vector (#46757, Addgene) using the mMESSAGE mMACHINE T3 Kit (#AM1348, Thermo Fisher Scientific). sgRNA (30 pg) and nls-zCas9-nls mRNA (150 pg) were injected into one-cell stage zebrafish embryos. The *reck*[ulb3] mutant allele harbors a 12 base pair deletion in the exon that encodes part of the third cysteine knot motif. Please refer to *Figure 1—figure supplement 2* for more information. Genotyping of *reck*[ulb3] mutant fish was performed by HRMA of PCR products using the following primers:

> *reck ulb3* forward: 5' – TTACGCAGGCAGACACACCACCTG – 3'
> *reck ulb3* reverse: 5' – GAGACGGTGGGAGACGAGTCTGTG – 3'

## High-resolution melt analysis

A CFX96 Touch Real-Time PCR Detection System (Bio-Rad) was used for PCRs and subsequent HRMA. Precision melt supermix for HRMA (Bio-Rad) was used in these experiments. PCR protocols were: 95°C for 2 min, 46 cycles of 95°C for 10 s, and 60°C for 30 s. Following the PCR, a high-resolution melt curve was generated by collecting EvaGreen fluorescence data in the 65–95°C range. The analyses were performed on normalized derivative plots.

## Heatshock treatments

Fish embryos/larvae raised at 28.5°C were subjected to a heatshock at 37°C for 1 hr by replacing egg water with pre-warmed (37°C) egg water, as described previously (*Matsuoka et al., 2016*; *Matsuoka*

*et al., 2017*). After each heatshock, the fish embryos/larvae were kept at room temperature (RT) for 10 min to cool down and then incubated at 28.5°C. Heatshock experiments were performed as follows: *Tg(kdrl:EGFP)^{s843};Tg(kdrl:NLS-mCherry)^{is4}* reporters-carrying *Tg(hsp70l:sflt1, cryaa-cerulean)^{bns80}*, *Tg(hsp70l:sflt4, cryaa-cerulean)^{bns82}*, and their sibling animals without the *cryaa*:cerulean eye marker, were subjected to a heatshock at 34, 44, and 54 hpf, and imaged at 72 or 144 hpf, for vascular analysis in the NH/OVLT brain regions. For analysis of CVP formation, the same Tg lines were employed and subjected to either a heatshock at 34, 44, and 54 hpf and imaged at 72 hpf or a heatshock at 34, 44, 54, 78, 102, 126 hpf and imaged at 144 hpf.

## Immunohistochemistry

Immunohistochemistry was performed by following standard immunostaining procedures as described previously (*Parab et al., 2021*), except CVP immunostaining and immunohistochemistry using anti-Pgp and anti-Glut1 antibodies. Both immunostaining procedures are described in detail as follows. For CVP analysis of 72 hpf or 6 dpf larvae, fish were fixed in pH adjusted (pH 7.0) 4% paraformaldehyde (PFA)/phosphate buffered saline (PBS) overnight at 4°C and dehydrated through immersion in methanol serial dilutions (50%, 75%, then 100% methanol three times, 10 min each) at RT. The dehydrated samples were stored in 100% methanol at –20°C before eye dissections. Microdissections of only one eye per fish was carried out under SMZ18 stereomicroscope (Nikon) at ×5–8 magnification using Dumont forceps. Dissected eyes were collected for immunostaining while the remaining tissues were collected for genotyping when necessary. The samples were rehydrated through immersion in methanol serial dilutions (50%, then 25% methanol, 10 min each) at RT and washed briefly three times in 1% PBST (1% Triton X-100 in 0.1 M PBS) followed by permeabilization in 10 µg/mL Proteinase K in 1% PBST at RT for 15 min. Samples were blocked at RT for 2–4 hr in the blocking solution containing 0.5% bovine serum albumin in 1% PBST prior to primary antibody incubation.

For immunohistochemistry using anti-Pgp and anti-Glut1 antibodies, overnight PFA fixed samples were dehydrated in the same manner as described earlier. For better visualization of ventral brain regions where the NH and OVLT are located, both the jaws and dorsal part of the brain were removed using forceps. No dissection was performed for samples that required the visualization of dorsal brain regions, including the dCP and mCP. After rehydration through immersion in methanol serial dilutions (50%, 25%, then 1% Triton-X/PBS), both dissected and undissected samples were incubated with 3% $H_2O_2$ in water for 40 min at RT followed by 100% acetone treatment at 4°C for 8 min. The samples were briefly washed in 1% PBST twice at RT and incubated in the blocking solution for 2–4 hr.

The following primary antibodies were used: chicken anti-GFP (GFP-1010, Aves Labs, 1:1000), rabbit anti-DsRed (#632496, Clontech, 1:300), mouse anti-Claudin-5 (#35-2500, Thermo Fisher Scientific, 1:500), mouse anti-ZO-1 (#33-9100, Thermo Fisher Scientific, 1:500), mouse anti-Pgp (#MA1-26528, Thermo Fisher Scientific, 1:100), and rabbit anti-Glut1 (#NB300-666, Novus Biologicals, 1:200). After primary antibody incubation at 4°C overnight, all samples were washed, incubated with secondary antibodies, and processed for imaging, as described previously (*Parab et al., 2021*).

## in situ hybridization combined with immunostaining in zebrafish larvae

Whole-mount in situ hybridization was conducted by following the protocol as previously described (*Machluf and Levkowitz, 2011*). Briefly, digoxigenin (DIG) labeled RNA probes were prepared by reverse transcription and purification of the amplified partial coding sequence of selected genes using a DIG RNA labeling kit (Roche, Switzerland). For 75 hpf and 150 hpf larvae, 30 min and 40 min proteinase K digestion was conducted, respectively. Probes for chain hybridization reaction (HCR) in situ hybridization v3.0 were ordered from Molecular Instruments (USA). The protocol of HCR in situ hybridization on whole-mount zebrafish larvae was slightly modified from the previously described protocol (*Choi et al., 2016*; *Choi et al., 2018*). Eight pmol probes were used for each gene and samples were fixed in 4% PFA/PBS and washed between detection and amplification steps. Immunostaining with anti-eGFP (GFP Rabbit Polyclonal #A-11122 or GFP Chicken Polyclonal #A10262, Thermo Fisher Scientific) was performed after in situ hybridization as described previously (*Chen et al., 2020*). The larvae were then mounted in 75% glycerol and imaged with a Zeiss LSM980 confocal microscope. Single plane and serial z-stack images were extracted using Fiji-ImageJ software (Fiji, Japan).

## Plasmid injections

For *pomca:mScarlet* plasmid injection, the plasmid was generated by cloning the mScarlet coding sequence downstream of 1006 bp *pomca* regulatory elements (*Anbalagan et al., 2018*; *Liu et al., 2003*) flanked by Tol2 sites. The 1006 bp *pomca* regulatory elements were amplified by PCR from genomic DNA of zebrafish embryos using the following primers:

*pomca* promoter forward: 5' – atta<u>aagctt</u>CTTTTTTATTCTGCTTTAAGACCTC – 3'
*pomca* promoter reverse: 5' – atta<u>gaattc</u>CTCTGAGAACACAATACAATTCAC – 3'

HindIII and EcoRI restriction enzyme sites (underlined) were incorporated into the primers to allow cloning. The engineered construct (30 pg per embryo) was injected together with transposase mRNA (*tol2* mRNA, 25 pg per embryo) into *Tg(UAS:EGFP-CAAX)^m1230* one-cell stage embryos that carry either the *TgBAC(vegfab:gal4ff)^bns273* or *TgBAC(vegfaa:gal4ff)^lri96* transgene.

## Dye tracer injections and quantification of tracer permeability

Tetramethylrhodamine-conjugated 3,000 MW (D3308, Thermo Fisher Scientific) and Alexa Fluor 647-conjugated 10,000 MW (D22914, Thermo Fisher Scientific) dextran dyes were co-injected into the common cardinal vein of 6 dpf *Tg(kdrl:EGFP)^s843* larvae. Injected live fish were imaged after approximately 30 min to 1 hr. Tracer permeability quantification was performed in a similar manner as previously described (*O'Brown et al., 2019*). Specifically, dye intensity within both the parenchyma and vasculature was measured on *z*-stack maximum projection images using the ImageJ software. Since each injected fish displayed slightly different intensity of circulating tracer, the intensity of the parenchyma was normalized by the tracer intensity within the vasculature in the same animal to account for differential amounts of circulating tracer. Measurements were carried out using the polygon selection tool in ImageJ.

### Midbrain parenchyma

In each fish, six paired sets of tracer intensity within the parenchyma and vasculature were measured in the midbrain area where a functional BBB is established by 5 dpf (*O'Brown et al., 2019*). These six paired sets were randomly selected, and tracer intensity measurements within vessels were carried out in lengths of approximately 20–60 µm with approximately 40 µm being most common. Parenchyma intensity measurements were conducted in an area directly adjacent to the corresponding vessel with approximately equal length and width. Each parenchyma intensity measurement was normalized by the paired vessel intensity measurement, then all six of these normalized intensity measurements were averaged to represent the tracer intensity in the midbrain parenchyma of each animal.

### dCP

Tracer intensity within each side of bilateral PGA vessels was measured and then averaged. Single parenchyma intensity measurement was obtained for the entire area enclosed by the PrA, PGV, and ACeV vessels, then normalized by the averaged tracer intensity within the PGA vasculature.

### Hy loop

The Hy loop was divided bilaterally, and tracer intensity within each side of the vascular loop was measured and then averaged. The tracer intensity of the entire parenchymal area enclosed by the Hy loop was measured, then normalized by the averaged tracer intensity within the Hy loop vasculature.

### OVLT

Tracer intensity was measured within the rostral part of the HyA that is anticipated to sit in close proximity to the OVLT (*García-Lecea et al., 2017*). Vessel lengths of approximately 20–40 µm were used to measure tracer intensity within the HyA. The parenchyma tracer intensity was measured in bilateral areas directly adjacent to the HyA with approximately equal size and then averaged. The averaged parenchyma tracer intensity was normalized by the tracer intensity within the HyA.

## Confocal and stereomicroscopy

A Leica TCS SP8 confocal laser scanning microscope (Leica) was used for most live, immunofluorescence, and transmitted differential interference contrast imaging. Fish embryos and larvae were

anesthetized with a low dose of tricaine, embedded in a layer of 1% low melt agarose in a glass-bottom Petri dish (MatTek), and imaged using a 25× water immersion objective lens. Leica Application Suite X (LAS X) software (Version 3.7.0.20979) was used for image acquisition and analysis. A Zeiss LSM980 confocal microscope (Zeiss) was used for imaging of samples subjected to in situ hybridization. ZEN 2.3 blue edition software was used for image acquisition and analysis using this microscope.

### Transmission electron microscopy

Wild-type larvae at 10 dpf were anesthetized with tricaine and fixed by immersion in 2.5% glutaraldehyde/4% PFA/0.2 M sodium cacodylate (pH 7.4) for 2 days at 4°C. Samples were post-fixed in 1% osmium tetroxide for 1 hr and incubated in 1% uranyl acetate in maleate buffer for 1 hr. Samples were dehydrated through immersion in methanol serial dilutions and embedded in epoxy resin. Ultrathin sections of 85 nm were cut with a diamond knife, collected on copper grids, and stained with uranyl acetate and lead citrate. Images were captured using a transmission electron microscope (FEI Tecnai G2 Spirit BioTWIN).

### Quantification of DLV, PCeV, MCeV, and MsV formation

Fish carrying the *Tg(kdrl:EGFP)*$^{s843}$ reporter were used for this quantification. Fish larvae at indicated developmental stages were analyzed for the presence or absence of the DLV. To quantify bilaterally formed PCeV, MCeV, and MsV, the following criteria were used to score the extent of each vessel's formation: (1) score 2 – when the bilateral vessels are both fully formed; (2) score 1.5 – when the vessel on one side is fully formed, but that on the other side is partially formed; (3) score 1 – when bilateral vessels are both partially formed or when the vessel on one side is fully formed, but that on the other side is absent; (4) score 0.5 – when the vessel on one side is partially formed, but that on the other side is absent; and (5) score 0 – when the bilateral vessels are both absent.

### Quantification of PrA, PGV, and ACeV formation

Fish carrying the *Tg(kdrl:EGFP)*$^{s843}$ reporter were used for this quantification. Fish larvae were analyzed at 72 hpf or 10 dpf for the extent of vascular formation at the dCP/PG interface. To quantify bilaterally formed PrA, PGV, and ACeV, the same scoring criteria used to quantify the extent of bilateral vessel formation (PCeV, MsV, and MCeV) described earlier were applied.

### Quantification of endothelial cells that comprise the CVP

Fish carrying *Tg(kdrl:EGFP)*$^{s843}$; *Tg(kdrl:NLS-mCherry)*$^{is4}$ or *Tg(kdrl:EGFP)*$^{s843}$ reporter were used to quantify the number of endothelial cells that formed the CVP. Fish larvae were fixed at 72 hpf or 6 dpf, dehydrated through methanol serial dilutions, subjected to eye dissections, and then immunostained for GFP, DsRed, and ZO-1, a marker for RPE. Immunostained eyes were imaged from the back of the eyes to capture the CVP that resides outside of the RPE layer. *Tg(kdrl:*EGFP)$^+$ or *Tg(kdrl:*NLS-mCherry)$^+$ cells that resided outside of the RPE layer and that located directly adjacent to RPE were defined and counted as CVP endothelial cells.

### Quantification of HyA, Hy loop, and HyV

Fish carrying the *Tg(kdrl:EGFP)*$^{s843}$ reporter were used for this quantification. Fish larvae were analyzed at 72 or 120 hpf for the extent of hypophyseal vascular formation. To quantify bilaterally formed HyV, the same scoring criteria used to quantify the extent of bilateral vessel formation (PCeV, MsV, and MCeV) described earlier were applied. HyA formation was quantified as follows: (1) score 1 – when the entire vessel is fully formed with vessel extension from the Hy loop to the PLA; (2) score 0.5 – when the vessel formation is partial and failed to reach the PLA, but passed the mid-point between HyA connection sites with the Hy loop and with the PLA; and (3) score 0 – either complete absence of the vessel or when the vessel formation is partial, failing to reach the mid-point between HyA connection sites with the Hy loop and with the PLA. The Hy loop was quantified as follows: (1) score 1 – when the entire capillary loop is fully formed; (2) score 0.5 – when capillary loop formation is partial, lacking a part of the capillary loop; and (3) score 0 – when the loop is completely absent.

### Quantification of endothelial cells that comprise the HyA, Hy loop, and HyV

Fish carrying both *Tg(kdrl:EGFP)*$^{s843}$ and *Tg(kdrl:NLS-mCherry)*$^{is4}$ reporters were used to quantify the number of endothelial cells that formed the HyA, Hy loop, and HyV. Cell quantification was performed

by manual counting of endothelial cell nuclei marked by the *Tg(kdrl:NLS-mCherry)* reporter. The bilateral sides of the HyV were used for endothelial cell counting from the midline up to the lateral brain region where the HyV and the cranial division of the internal carotid artery crossed on *z*-stack maximum projection images. Endothelial cell counting of the HyA was conducted on the vessel extension between HyA connection sites with the Hy loop and with the PLA.

## Statistical analysis

Statistical differences for mean values among multiple groups were determined using a one-way analysis of variance (ANOVA) followed by Tukey's multiple comparison test. Fisher's exact test was used to determine significance when comparing the degree of penetrance of observed phenotypes. Statistical analyses were performed using GraphPad Prism 8.1.1. The criterion for statistical significance was set at $p < 0.05$. Error bars are SD.

## Acknowledgements

We thank Drs. Didier Stainier, Nathan Lawson, and Michael Taylor for kindly providing us with fish lines; Dr. Natasha O'Brown for advice on dye tracer quantification; Dr. Isha Goel for testing immunostaining conditions; Don Zeisloft and his team for zebrafish care and husbandry; Leslie Sanderson for assistance of zebrafish mutant maintenance; and Dr. Judith Drazba and her team for confocal and electron microscopy imaging.

## Additional information

### Funding

| Funder | Grant reference number | Author |
| --- | --- | --- |
| National Institutes of Health | R01 NS117510 | Ryota L Matsuoka |
| Cleveland Clinic Foundation | | Ryota L Matsuoka |
| Israel Science Foundation | #349/21 | Gil Levkowitz |
| Hedda, Alberto, and David Milman Baron Center for Research on the Development of Neural Networks | | Gil Levkowitz |

The funders had no role in study design, data collection and interpretation, or the decision to submit the work for publication.

### Author contributions

Sweta Parab, Data curation, Formal analysis, Validation, Investigation, Visualization, Writing - original draft; Olivia A Card, Qiyu Chen, Data curation, Formal analysis, Validation, Investigation, Visualization; Michelle America, Data curation, Formal analysis, Investigation, Validation, Visualization; Luke D Buck, Rachael E Quick, Formal analysis, Investigation; William F Horrigan, Investigation; Gil Levkowitz, Resources, Funding acquisition, Supervision; Benoit Vanhollebeke, Resources, Supervision, Validation; Ryota L Matsuoka, Conceptualization, Resources, Data curation, Formal analysis, Supervision, Funding acquisition, Validation, Investigation, Visualization, Writing - original draft, Project administration, Writing – review and editing

### Author ORCIDs

Sweta Parab ⬤ http://orcid.org/0000-0002-9932-5117
Gil Levkowitz ⬤ http://orcid.org/0000-0002-3896-1881
Benoit Vanhollebeke ⬤ http://orcid.org/0000-0002-0353-365X
Ryota L Matsuoka ⬤ http://orcid.org/0000-0001-6214-2889

### Ethics

This study was performed in strict accordance with the recommendations in the Guide for the Care and Use of Laboratory Animals of the National Institutes of Health. All zebrafish husbandry was performed under standard conditions in accordance with institutional and national ethical and animal welfare guidelines. All zebrafish work was approved by the Cleveland Clinic's Institutional Animal Care and Use Committee under the protocol number 00002684. Every effort was made to minimize suffering and distress of the animals used throughout this study.

### Decision letter and Author response

Decision letter https://doi.org/10.7554/eLife.86066.sa1
Author response https://doi.org/10.7554/eLife.86066.sa2

---

## Additional files

### Supplementary files
• MDAR checklist

### Data availability

All data generated or analyzed during this study are included in the manuscript and supporting file; Source data files have been provided for Figures 9 and 10.

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
