## [Editor Report]

This study presents a fundamental approach to understanding genetic redundancy of Vegf family, Wnt7, Gpr124 and Reck in mediating zebrafish blood brain barrier angiogenesis and formation of region-specific fenestrated capillaries. The work comprehensively examines interplay of Vegfc/d and Vegfa and their sources for regionally-restricted angiogenesis. The study provides detail on the region-specific expression of Vegf ligands, and assesses the permeability of the fenstrated capillaries using dye injections into the vasculature.

---

## [Decision Letter]

**Decision letter after peer review:**

Thank you for submitting your article "Local angiogenic interplay of Vegfc/d and Vegfa drives brain region-specific development of fenestrated capillaries" for consideration by *eLife*. Your article has been reviewed by 3 peer reviewers, and the evaluation has been overseen by Joseph Gleeson as the Reviewing Editor and Marianne Bronner as the Senior Editor. The following individual involved in the review of your submission has agreed to reveal their identity: Michael R Taylor (Reviewer #3).

Essential revisions:

1. Authors should more carefully discuss the expression of the VEGF receptors, ideally documenting specific expression domains in the manuscript.

2. The issue of capillary fenestration was raised by several reviewers. Authors should more carefully describe the definition and consider functionally interrogating the aspect of fenestration because the plvap:EGFP marker could be non-specific.

*Reviewer #1 (Recommendations for the authors):*

The manuscript of Parab et al. reports a beautiful phenotype analysis of the vascular brain/meningeal anatomy in a variety of reporter lines and mutants for Wnt/β-catenin signaling and angiogenic cues (Vegfaa, Vegfab Vegfc, Vegfd) during zebrafish development.

The present study extends the previous work of the same Parab, Quick, and Matsuoka, that focused on fenestrated vessel formation in the zebrafish myelencephalic choroid plexus (mCP). Vegfs were shown to regulate fenestrated vessel formation in combination, but not individually, and with only little effect on neighboring non-fenestrated brain vessel development. The fenestrated endothelium is thus known to have specific angiogenic requirements.

The scale of investigation has now changed, and fenestrated vessel formation has been examined throughout the brain, in both circumventricular organs (organum vasculosum of lamina terminalis) and other choroid plexuses (CPs) including the diencephalic CP and its interface with the pineal gland, the eye choroid (choriocapillaris), and the hypophysis vasculature. The original finding is that a region-specific code of angiogenic cues controls fenestrated vessel formation. The authors show that fenestrated vessels form independently of Wnt/β-catenin signaling and BBB vascular development but require different combinations of Vegfa and Vegfc/d-dependent angiogenesis within and across brain regions. A previously unappreciated function of autocrine and paracrine Vegfc signaling is demonstrated in this brain region-specific regulation of fenestrated capillary development.

Twenty-one different fish lines accurately genotyped and characterized and including a new Reck mutant, have been instrumental to conduct vascular pattern analysis, using confocal and stereomicroscopy imaging combined with transmission EM. High-quality illustration and robust quantification methods, previously validated, have been used. The study is well organized and reflects the high expertise and strong methodology of the investigators. Data are presented in nine dense figures and the contribution of angiogenic ligands to fenestrated vessel formation can hardly be studied more in-depth.

However, and this will be my only main concern, no information is provided on the regional diversity of angiogenic receptor expression that may correlate with the regional angiogenic factor code. Without asking for a spatial transcriptomic study, the combination of Vegfr-reporter lines or in situ hybridization with a combination of receptor probes would allow for generating a comprehensive set of ligand/receptor data relative to the regional angiogenic signaling pattern involved in fenestrated vessel formation.

*Reviewer #2 (Recommendations for the authors):*

1. In Figure 1, the red staining for panels C-F' was not labeled. Accompanying the quantification data of H-N, it might be helpful to add a panel to quantify the disrupted parenchymal angiogenesis in these Wnt signaling mutants.

2. In all the figures, it might be easier for readers to follow, if the panels would be presented in the order of Vegfaa, Vegfab, Vegfc, Vegfd, and their combinations, rather than the current order based on phenotype severity.

3. It might be easier for readers to follow if the data on fenestrated capillary development presented in current Fig7-9 would be moved right after figure 4. In this case, figure 1 introduced dorsal (dCP and mCP) and ventral (OVLT and NH) views of fenestrated vessel structures, and the following figures investigate the details in this order. The data presented in current Figure 5 and 6 on retinal choriocapillaris disrupts the flow and make it a bit convoluted to read through. Authors can also consider moving these two figures to supplemental or condensed to one figure.

*Reviewer #3 (Recommendations for the authors):*

1) There is no examination or discussion of the Vegf receptors. Similar to the examination of vegf ligands in this study, previous work has shown regional differences in vegf receptor expression during development (for example, see Volgrin 2019 Cell Reports). Thus, a description of specific vegf receptor expression (kdr, kdrl, and flt4) and how this correlates with the expression of vegf ligands and vegf mutants in the current study could strengthen the manuscript. In addition, there is no mention of the zebrafish vegf ligands vegfba and vegfbb.

2) Previous work has shown that Plvap expression is upregulated in response to Vegf signaling but does not necessarily indicate the presence of fenestrations. While the current study demonstrates the requirement of specific combinations of vegf ligands for brain region-specific angiogenesis, the science could be strengthened by examining the expression of plvap:EGFP in the various vegf mutants, potentially indicating which individual or combination of vegf ligands drive Plvap expression in these vessels.

3) For the quantification of blood vessels in which vessel formation was scored from 0-2, there is no indication of how partially formed vessels on either side of the brain would be scored, unless both sides always looked similar. Could this quantification be explained more clearly?

4) For improved clarity, this reviewer suggests adding descriptions to the figure legends of the vessel schematics for the specific development timepoint (or range) that the schematic is representing.

5) In the legend for Figure 2, the lateral view M' appears to have been misidentified as N'.

6) For images representing phenotypes that are not fully penetrated (and similarly for the schematics in Figure 4K), are these images representative of the average phenotype, the most common phenotype, or the most severe phenotype?

7) The legend for Figures 6L and M is missing the number of animals per genotype for those graphs.

8) The combination of green and red fluorescence in merged images is not visible to many colorblind individuals. Therefore, for improved accessibility, it is suggested that the authors modify images, especially when only the merge is displayed.

9) Graphs in Figure 6 would benefit from information on the significance between different phenotypes. Specifically, in Figure 6E, groups 7 and 8 are not compared to groups 5 and 6 (respectively), despite this being part of the claim that increased deletion of either vegfa gene resulted in reduced numbers, and in R groups 3, 4, and 6 are not compared to WT.

10) Several references are missing or not completely accurate.

a. In the Introduction (Genetic loss…second paragraph), "…CPs serve as a gateway for immune cells from the circulatory system to the brain." No references are provided.

b. In Results (Genetic loss…first paragraph), the plvap:EGFP and glut1b:mCherry transgenic fish are not referenced. Furthermore, as described above, plvap:EGFP does not necessarily "mark fenestrated endothelial cells" but instead "labels plvap-positive cells."

c. In Results (second paragraph), "We observed that gpr124s984, reckulb3, or wnt7aaulb2…" there should be a reference for Ulrich et al. 2016 Development describing the recky72 mutant and not the unpublished ulb3 allele or at least clarify the writing.

d. In Results (Fenestrated vascular…first paragraph), references are missing for Claudin 5 expression in CP epithelial cells as this result has already been published (please see van Leeuwen et al. 2018 Biology Open; Henson et al. 2014 Front. Neurosci.; others?).

e. In Results (Normal choriocapillaris…first sentence), references are missing.

---

## [Author Response]

Essential revisions:1. Authors should more carefully discuss the expression of the VEGF receptors, ideally documenting specific expression domains in the manuscript.

We have now included a paragraph discussing potential contributions of the Vegf receptors to CP and CVO vascularization based on the reported biochemical, expression, and genetic studies on zebrafish Vegf receptors. Our ongoing genetic mutant analysis implies that multiple Vegf receptors are involved redundantly in CP and CVO vascularization. Thus, we anticipate that a careful and thorough analysis of their expression patterns and functional redundancy will be needed at the comparable levels to what has been investigated in this study with respect to Vegf ligands. To date, a detailed expression analysis of all four Vegf receptors during CP and CVO vascularization is not available from literature and will likely be best addressed with new tools combined with optimizations of existing protocols. Nevertheless, as stated above, we provide a new paragraph discussing our perspectives on potential Vegf receptor involvement in CP and CVO vascularization in the revised version of the manuscript.

2. The issue of capillary fenestration was raised by several reviewers. Authors should more carefully describe the definition and consider functionally interrogating the aspect of fenestration because the plvap:EGFP marker could be non-specific.

We have now addressed this concern by conducting several new experiments, including vascular permeability analysis using dye tracers of different molecular weights and immunolabeling with additional endothelial cell markers. In the first set of experiments shown in new Figure 1C–G, 2C–D”, and Figure 1—figure supplement 1A–F”, we provide evidence that two well-established BBB markers, Glut1 and P-glycoprotein (PGP), are co-localized with *Tg(glut1b:mCherry)* transgene expression in brain vasculature, while both proteins are undetectable in strong *Tg(plvap:EGFP)^+^* blood vessels that form in the CPs and CVOs. We also examined β-catenin activities across different brain vascular beds using the previously established endothelial cell-specific β-catenin activity reporter line and observed reduced expression of the β-catenin reporter in strong *Tg(plvap:EGFP)^+^* brain vasculature (Figure 1figure supplement 1G–K). Finally, we assessed brain vascular permeability by injecting dye tracers into the bloodstream via the common cardinal vein and observed significantly higher tracer permeability around the CP and CVO regions than that detected in the midbrain regions where the BBB forms (Figure 2L–Q). Combined with the strong correlation between robust *Tg(plvap:EGFP)* reporter expression and detection of endothelial fenestration at the ultrastructure level (the choriocapillaris in Figure 10A–D of this study and the neurohypophysis in Figure 2B of this study and the published study by Gordon et al., Development, 146 (23):dev177790, 2019), we believe that these additional experiments sufficiently address the concern of the *Tg(plvap:EGFP)* reporter specificity and clarify fenestrated endothelial cell molecular signatures and properties in the zebrafish CP and several CVOs, similarly to what has been observed in the mammalian counterparts.

Reviewer #1 (Recommendations for the authors):The manuscript of Parab et al. reports a beautiful phenotype analysis of the vascular brain/meningeal anatomy in a variety of reporter lines and mutants for Wnt/β-catenin signaling and angiogenic cues (Vegfaa, Vegfab Vegfc, Vegfd) during zebrafish development.The present study extends the previous work of the same Parab, Quick, and Matsuoka, that focused on fenestrated vessel formation in the zebrafish myelencephalic choroid plexus (mCP). Vegfs were shown to regulate fenestrated vessel formation in combination, but not individually, and with only little effect on neighboring non-fenestrated brain vessel development. The fenestrated endothelium is thus known to have specific angiogenic requirements.The scale of investigation has now changed, and fenestrated vessel formation has been examined throughout the brain, in both circumventricular organs (organum vasculosum of lamina terminalis) and other choroid plexuses (CPs) including the diencephalic CP and its interface with the pineal gland, the eye choroid (choriocapillaris), and the hypophysis vasculature. The original finding is that a region-specific code of angiogenic cues controls fenestrated vessel formation. The authors show that fenestrated vessels form independently of Wnt/β-catenin signaling and BBB vascular development but require different combinations of Vegfa and Vegfc/d-dependent angiogenesis within and across brain regions. A previously unappreciated function of autocrine and paracrine Vegfc signaling is demonstrated in this brain region-specific regulation of fenestrated capillary development.Twenty-one different fish lines accurately genotyped and characterized and including a new Reck mutant, have been instrumental to conduct vascular pattern analysis, using confocal and stereomicroscopy imaging combined with transmission EM. High-quality illustration and robust quantification methods, previously validated, have been used. The study is well organized and reflects the high expertise and strong methodology of the investigators. Data are presented in nine dense figures and the contribution of angiogenic ligands to fenestrated vessel formation can hardly be studied more in-depth.However, and this will be my only main concern, no information is provided on the regional diversity of angiogenic receptor expression that may correlate with the regional angiogenic factor code. Without asking for a spatial transcriptomic study, the combination of Vegfr-reporter lines or in situ hybridization with a combination of receptor probes would allow for generating a comprehensive set of ligand/receptor data relative to the regional angiogenic signaling pattern involved in fenestrated vessel formation.

We appreciate this reviewer’s positive and encouraging comments highlighting both the quality and significance of our study. As we commented in response to the Essential Revisions point #1, we anticipate that a detailed expression analysis of all four Vegf receptors at different developmental stages during CP and CVO vascularization will be best addressed with new technologies combined with optimizations of existing tools/protocols. Thus, we have provided a paragraph of discussion on our perspectives for potential Vegf receptors involved in CP and CVO vascularization in the current study.

Reviewer #2 (Recommendations for the authors):1. In Figure 1, the red staining for panels C-F' was not labeled. Accompanying the quantification data of H-N, it might be helpful to add a panel to quantify the disrupted parenchymal angiogenesis in these Wnt signaling mutants.

We appreciate these suggestions and have now added color-coded *z*-depth information next to Figure 1 panels I-L’. Regarding the second point, we fully agree with the reviewer that adding a quantification panel of parenchymal angiogenesis will be helpful. However, BBB parenchymal angiogenesis defects in these mutants were already well characterized and documented in previous studies that are referenced in our revised manuscript. Thus, we have cited these references to allow readers to obtain detailed information from these published studies.

2. In all the figures, it might be easier for readers to follow, if the panels would be presented in the order of Vegfaa, Vegfab, Vegfc, Vegfd, and their combinations, rather than the current order based on phenotype severity.

We appreciate this suggestion. One of the main questions in this study, as stated in the last paragraph of the Introduction, was to address whether a combination of Vegf ligands (Vegfab, Vegfc, and Vegfd) crucial for mCP vascularization is a universal angiogenic inducer of fenestrated capillary formation across the brain. This study flow led us to present these three *vegf* mutant results first in most parts. Thus, we would like to keep this study flow as it is, although we agreed with the reviewer that it might be easier for some readers to follow in the order of Vegfaa, Vegfab, Vegfc, and Vegfd.

3. It might be easier for readers to follow if the data on fenestrated capillary development presented in current Fig7-9 would be moved right after figure 4. In this case, figure 1 introduced dorsal (dCP and mCP) and ventral (OVLT and NH) views of fenestrated vessel structures, and the following figures investigate the details in this order. The data presented in current Figure 5 and 6 on retinal choriocapillaris disrupts the flow and make it a bit convoluted to read through. Authors can also consider moving these two figures to supplemental or condensed to one figure.

We appreciate this great suggestion and agree with the reviewer that presenting Fig7-9 after Figure 4 in our original manuscript will make the flow better. We have now moved these figures as suggested by the reviewer in the revised version of the manuscript.

Reviewer #3 (Recommendations for the authors):1) There is no examination or discussion of the Vegf receptors. Similar to the examination of vegf ligands in this study, previous work has shown regional differences in vegf receptor expression during development (for example, see Volgrin 2019 Cell Reports). Thus, a description of specific vegf receptor expression (kdr, kdrl, and flt4) and how this correlates with the expression of vegf ligands and vegf mutants in the current study could strengthen the manuscript. In addition, there is no mention of the zebrafish vegf ligands vegfba and vegfbb.

We appreciate this great suggestion and have now added two paragraphs of discussions that address these points raised by the reviewer in the revised manuscript.

2) Previous work has shown that Plvap expression is upregulated in response to Vegf signaling but does not necessarily indicate the presence of fenestrations. While the current study demonstrates the requirement of specific combinations of vegf ligands for brain region-specific angiogenesis, the science could be strengthened by examining the expression of plvap:EGFP in the various vegf mutants, potentially indicating which individual or combination of vegf ligands drive Plvap expression in these vessels.

We appreciate this important suggestion and completely agree with the reviewer that examining *Tg(plvap:EGFP)* expression and/or fenestration states in the various *vegf* mutants we have analyzed in this manuscript will significantly strengthen this study and be the crucial next step. However, detailed qualitative and quantitative analyses of these endothelial states in single, double, and triple *vegf* mutants requires substantial time and manpower for fish crosses and careful analyses, which we believe is beyond the scope of the current study.

3) For the quantification of blood vessels in which vessel formation was scored from 0-2, there is no indication of how partially formed vessels on either side of the brain would be scored, unless both sides always looked similar. Could this quantification be explained more clearly?

We appreciate this comment. In the original manuscript, we provided explanations as to vessel formation scoring from 0-2, including partially formed vessels that is scored as 0.5 on either side. However, we have revised the descriptions on this scoring method. Please refer to the “Quantification of DLV, PCeV, MCeV, and MsV formation” descriptions in the Method of the revised manuscript.

4) For improved clarity, this reviewer suggests adding descriptions to the figure legends of the vessel schematics for the specific development timepoint (or range) that the schematic is representing.

We appreciate this great suggestion and have now added developmental timepoint or range to the figure legend of vascular schematics in the revised manuscript.

5) In the legend for Figure 2, the lateral view M' appears to have been misidentified as N'.

Thank you for pointing out this mislabeling. We have now corrected this mislabeling in the revised version of the manuscript.

6) For images representing phenotypes that are not fully penetrated (and similarly for the schematics in Figure 4K), are these images representative of the average phenotype, the most common phenotype, or the most severe phenotype?

We appreciate this comment. As the reviewer mentioned, our observed vascular phenotypes in most mutants are partially penetrant with variability. We tried our best to select each mutant image that well represents the quantification and statistical results. These images are categorized as severe phenotypes in most cases.

7) The legend for Figures 6L and M is missing the number of animals per genotype for those graphs.

We appreciate this comment and have now added the number of animals to the figure legend of these graphs.

8) The combination of green and red fluorescence in merged images is not visible to many colorblind individuals. Therefore, for improved accessibility, it is suggested that the authors modify images, especially when only the merge is displayed.

We appreciate this suggestion and have now changed the color of red fluorescence into magenta in most merged images where the separate visualization of green and red fluorescent reporters is particularly important.

9) Graphs in Figure 6 would benefit from information on the significance between different phenotypes. Specifically, in Figure 6E, groups 7 and 8 are not compared to groups 5 and 6 (respectively), despite this being part of the claim that increased deletion of either vegfa gene resulted in reduced numbers, and in R groups 3, 4, and 6 are not compared to WT.

We appreciate this suggestion. We have now added the statistical information to these graphs and provided brief descriptions on these comparisons in the Results section of the revised manuscript.

10) Several references are missing or not completely accurate.a. In the Introduction (Genetic loss…second paragraph), "…CPs serve as a gateway for immune cells from the circulatory system to the brain." No references are provided.

We have revised this sentence and added references in the revised manuscript.

b. In Results (Genetic loss…first paragraph), the plvap:EGFP and glut1b:mCherry transgenic fish are not referenced. Furthermore, as described above, plvap:EGFP does not necessarily "mark fenestrated endothelial cells" but instead "labels plvap-positive cells."

We appreciate this suggestion and have now added appropriate references in the revised version of the manuscript. We have also made significant revisions in describing the *Tg(plvap:EGFP)* reporter line.

c. In Results (second paragraph), "We observed that gpr124s984, reckulb3, or wnt7aaulb2…" there should be a reference for Ulrich et al. 2016 Development describing the recky72 mutant and not the unpublished ulb3 allele or at least clarify the writing.

We appreciate this comment and have now revised this sentence with the appropriate references in the revised version of the manuscript.

d. In Results (Fenestrated vascular…first paragraph), references are missing for Claudin 5 expression in CP epithelial cells as this result has already been published (please see van Leeuwen et al. 2018 Biology Open; Henson et al. 2014 Front. Neurosci.; others?).

We appreciate this suggestion and have now added these references in the revised version of the manuscript.

e. In Results (Normal choriocapillaris…first sentence), references are missing.

We appreciate this comment and have now added these references in the revised version of the manuscript.